# The SMC5/6 complex prevents genotoxicity upon APOBEC3A-mediated replication stress

Dylan F Fingerman[1,2,11], David R O'Leary [1,2,11], Ava R Hansen[1,2,3,11], Thi Tran[1,2], Brooke R Harris[1,2], Rachel A DeWeerd [1,2], Katharina E Hayer [4,5], Jiayi Fan[6], Emily Chen [6,7], Mithila Tennakoon [2,8], Alice Meroni[2,8], Julia H Szeto[4], Jessica Devenport[1,2], Danielle LaVigne[1,2], Matthew D Weitzman [4,9], Ophir Shalem [10], Jeffrey Bednarski [1,2], Alessandro Vindigni [2,8], Xiaolan Zhao[6] & Abby M Green [1,2 ✉]

## Abstract

**Mutational patterns caused by APOBEC3 cytidine deaminase activity are evident throughout human cancer genomes. In particular, the APOBEC3A family member is a potent genotoxin that causes substantial DNA damage in experimental systems and human tumors. However, the mechanisms that ensure genome stability in cells with active APOBEC3A are unknown. Through an unbiased genome-wide screen, we define the Structural Maintenance of Chromosomes 5/6 (SMC5/6) complex as essential for cell viability when APOBEC3A is active. We observe an absence of APOBEC3A mutagenesis in human tumors with SMC5/6 dysfunction, consistent with synthetic lethality. Cancer cells depleted of SMC5/6 incur substantial genome damage from APOBEC3A activity during DNA replication. Further, APOBEC3A activity results in replication tract lengthening which is dependent on PrimPol, consistent with re-initiation of DNA synthesis downstream of APOBEC3A-induced lesions. Loss of SMC5/6 abrogates elongated replication tracts and increases DNA breaks upon APOBEC3A activity. Our findings indicate that replication fork lengthening reflects a DNA damage response to APOBEC3A activity that promotes genome stability in an SMC5/6-dependent manner. Therefore, SMC5/6 presents a potential therapeutic vulnerability in tumors with active APOBEC3A.**

**Keywords** Replication Stress; Genome Integrity; Mutational Signatures; Cancer Mutagenesis; Cytidine Deaminase
**Subject Category** DNA Replication, Recombination & Repair

## Introduction

Cytidine deamination caused by APOBEC3 enzymes is among the most prevalent sources of endogenous mutagenesis in human cancers (Alexandrov et al, 2020; Burns et al, 2013b; Chan et al, 2015; Cortez et al, 2019; Jalili et al, 2020; Nik-Zainal et al, 2012; Roberts et al, 2013). APOBEC3 enzymes catalyze the conversion of cytidine to uracil in single-stranded (ss)DNA substrates, which can result in mutations after replication or uracil excision (Chen et al, 2006; Richardson et al, 2014). The APOBEC3 enzymes function in the innate immune system to deaminate and mutate viral genomes and retroelements to restrict infection and retrotransposition (Chen et al, 2006; Harris and Dudley, 2015; Richardson et al, 2014). Off-target or aberrant activity of the enzymes results in damage to the cellular genome (Baker et al, 2022; Burns et al, 2013a; Green et al, 2016; Haradhvala et al, 2016; Landry et al, 2011; Suspene et al, 2011; Venkatesan et al, 2021). Of the seven-member family (APOBEC3A-H), APOBEC3A is expressed in the nucleus and causes mutagenesis in experimental systems and human tumors, which can be genotoxic at high levels (Burns et al, 2013a; Cortez et al, 2019; DeWeerd et al, 2022; Petljak et al, 2022; Roberts et al, 2012). Mutational patterns of APOBEC3A activity are conserved across yeast and mammalian experimental models (Burns et al, 2013a; Chan et al, 2015; Hoopes et al, 2016; Law et al, 2020; Petljak et al, 2022; Roberts et al, 2012; Taylor et al, 2013).

In cancer, the genotoxic potential of APOBEC3A activity can be exploited by inhibition of the essential DNA damage responses which it activates. APOBEC3A deamination at replication forks activates replication stress responses initiated by ATR kinase signaling (Buisson et al, 2017; Green et al, 2017). Inhibition of ATR abrogates the cell cycle checkpoint, enables the accumulation of mutations during DNA replication, and ultimately promotes replication catastrophe as cells move through mitosis (Buisson et al, 2017; Green et al, 2017). Cytotoxicity of APOBEC3A activity upon ATR inhibition illustrates a synthetic lethal interaction and

[1]Department of Pediatrics, Washington University School of Medicine, St. Louis, MO, USA. [2]Center for Genome Integrity, Siteman Cancer Center, Washington University School of Medicine, St. Louis, MO, USA. [3]Drexel University College of Medicine, Philadelphia, PA, USA. [4]Division of Cancer Pathobiology, Children's Hospital of Philadelphia, Philadelphia, PA, USA. [5]Department of Biomedical and Health Informatics, Children's Hospital of Philadelphia, PA, USA. [6]Molecular Biology Program, Memorial Sloan-Kettering Cancer Center, New York, NY, USA. [7]School of Agriculture and Life Sciences, Cornell University, Ithaca, NY, USA. [8]Department of Medicine, Washington University School of Medicine, St. Louis, MO, USA. [9]Department of Pathology and Laboratory Medicine, University of Pennsylvania Perelman School of Medicine, Philadelphia, PA, USA. [10]Department of Genetics, University of Pennsylvania Perelman School of Medicine, Philadelphia, PA, USA. [11]These authors contributed equally: Dylan F Fingerman, David R O'Leary, Ava R Hansen. ✉E-mail: abby.green@wustl.edu

the essential nature of DNA damage responses in tumor cells undergoing mutagenesis. We employed the synthetic lethality strategy to investigate DNA damage responses that are elicited by the activity of APOBEC3A. Using a genome-wide CRISPR-based screen, we determined that the optimal function of the Structural Maintenance of Chromosomes 5/6 (SMC5/6) complex is essential in cells with active APOBEC3A.

SMC5/6 is a highly conserved eight-member complex comprised of SMC5 and SMC6 as well as six non-SMC element (NSMCE) proteins (Aragon, 2018). SMC5 and SMC6 dimers form the hinge-like backbone of the complex to which other subunits attach (Alt et al, 2017; Yu et al, 2022). Similar to the related condensin and cohesin SMCs, SMC5/6 interacts with DNA to influence genome stability. Notably, SMC5/6 can bind to both ssDNA and double-stranded (ds)DNA, and can stabilize ssDNA-dsDNA junctions (Chang et al, 2022; Tanasie et al, 2022). While condensin and cohesin act in chromosome folding and segregation, the function of SMC5/6 in genome maintenance is less well-defined.

Experimental suppression of SMC5/6, as well as germline defects in SMC5/6 in human syndromes result in replication and repair defects and chromosomal aberrations (Grange et al, 2022; Payne et al, 2014; van der Crabben et al, 2016; Venegas et al, 2020; Zhu et al, 2023). Despite these data indicating an important role for the complex in genome integrity, SMC5/6 deficiency in human cancer is poorly understood. In yeast and mammalian cells, SMC5/6 co-localizes with replication-associated proteins and nascent DNA, indicating that the complex localizes to replication structures (Alabert et al, 2014; Ampatzidou et al, 2006; Barlow et al, 2013; Betts Lindroos et al, 2006; Winczura et al, 2019). Several genome maintenance roles for SMC5/6 at replication forks have been elucidated, such as regulation of fork reversal and resolution of recombination intermediates that arise due to DNA repair at impaired replication forks (Chen et al, 2009; Irmisch et al, 2009; Potts and Yu, 2005; Wu et al, 2012). Additionally, SMC5/6 localizes to natural pausing sites at centromeres, telomeres, and ribosomal DNA even in the absence of genome stress, suggesting a role for support of replication through repetitive or fragile regions (Agashe et al, 2021; Barlow et al, 2013; Menolfi et al, 2015; Peng et al, 2018; Torres-Rosell et al, 2007). While the influence of SMC5/6 on replicating DNA is established, a role for the protection of forks undergoing cytidine deamination has not been defined.

In this study, we discovered a synthetic lethal interaction between APOBEC3A activity and loss of SMC5/6. By modeling SMC5/6 loss in cancer cell lines, we found that APOBEC3A activity elicited high levels of DNA breaks leading to genotoxic cell death. This synthetic lethal interaction was conserved from yeast to human tumors. In cancer cells depleted of SMC5/6, deaminase-induced DNA damage was maximal during DNA replication. Intriguingly, we found that APOBEC3A activity led to an increased length of replication forks as measured by DNA fiber imaging. The increased length was dependent on PrimPol, thus, is likely due to bypass of APOBEC3A-mediated replication obstacles by repriming downstream of a DNA lesion. Interestingly, the increased fork length was also dependent on SMC5/6. We propose a model in which SMC5/6 stabilizes replication forks in cells undergoing deaminase-mediated damage. These data demonstrate a new mechanism by which genome integrity is maintained in the context of APOBEC3A activity, and reveal a synthetic lethal interaction

that may provide opportunities for therapeutic targeting of SMC5/6 in cancer.

# Results

## Functional genomics screen identifies SMC5/6 as essential in cells with APOBEC3A activity

To identify cellular processes that ensure genome protection from the mutagenic activity of APOBEC3A, we employed a genome-wide functional screening approach. THP1 (myeloid leukemia) cells with integrated doxycycline (dox)-inducible APOBEC3A transgene (THP1-A3A) and constitutive Cas9 transgene (Appendix Fig. S1a) were transduced with the Brunello guide RNA (sgRNA) lentiviral library which includes multiple sgRNAs for each human gene as well as non-targeting control sgRNAs (Doench et al, 2016). A low lentivirus:cell ratio (MOI 0.4) was used to allow screening for knockout of each human gene independently within a pooled population (Appendix Fig. S1b). Following transduction, control cells (-dox) were cultured in parallel with cells induced to express APOBEC3A (+dox) for 15 days (Fig. 1A). In both groups, cells were harvested and sgRNAs were sequenced, normalized, and analyzed by three independent pipelines to generate a gene score for each gene represented in the Brunello library (Appendix Fig. S1c). Sequencing coverage of the entire library was similar across samples, regardless of dox treatment (Appendix Fig. S1d,e).

Comparison of APOBEC3A-expressing to control cells revealed under-represented or absent sgRNAs, indicating genes that were negatively selected (Dataset EV1). Negative selection is interpreted as cell death due to synthetic lethality between target gene loss and APOBEC3A expression. The top 250 negatively selected genes were analyzed for gene ontology which revealed biological processes clustered around DNA damage and repair, DNA and RNA metabolism, and chromosome organization (Fig. 1B). Among the negatively selected genes were ATR and CHEK1 (Fig. 1C), which have previously been demonstrated to be synthetically lethal with APOBEC3A expression (Buisson et al, 2017; Green et al, 2017). Furthermore, ATM sgRNA levels were unchanged, indicating no impact on the survival of APOBEC3A-expressing cells, consistent with prior findings (Buisson et al, 2017; Green et al, 2017) (Fig. 1C).

Within chromosome organization, which was the most significant GO term, SMC5 and NSMCE3, two members of the eight-protein complex comprising SMC5/6, were significantly negatively selected (Fig. 1C). Additional SMC5/6 genes were negatively selected, although they appeared lower on the list. To validate this potential synthetic lethal interaction, we depleted SMC5 in THP1-A3A cells and found that, upon APOBEC3A expression, cells were significantly less viable than controls (Fig. 1D). Thus, a functional SMC5/6 complex is essential for the viability of cells expressing APOBEC3A.

## SMC5/6 loss potentiates APOBEC3A-mediated genotoxicity

To explore the reproducibility of a synthetic lethal interaction between APOBEC3A and loss of SMC5/6, we depleted SMC5, SMC6, and/or NSMCE4 in cell lines from different tissues. SMC5/6 complex formation relies on all subunits being intact, thus

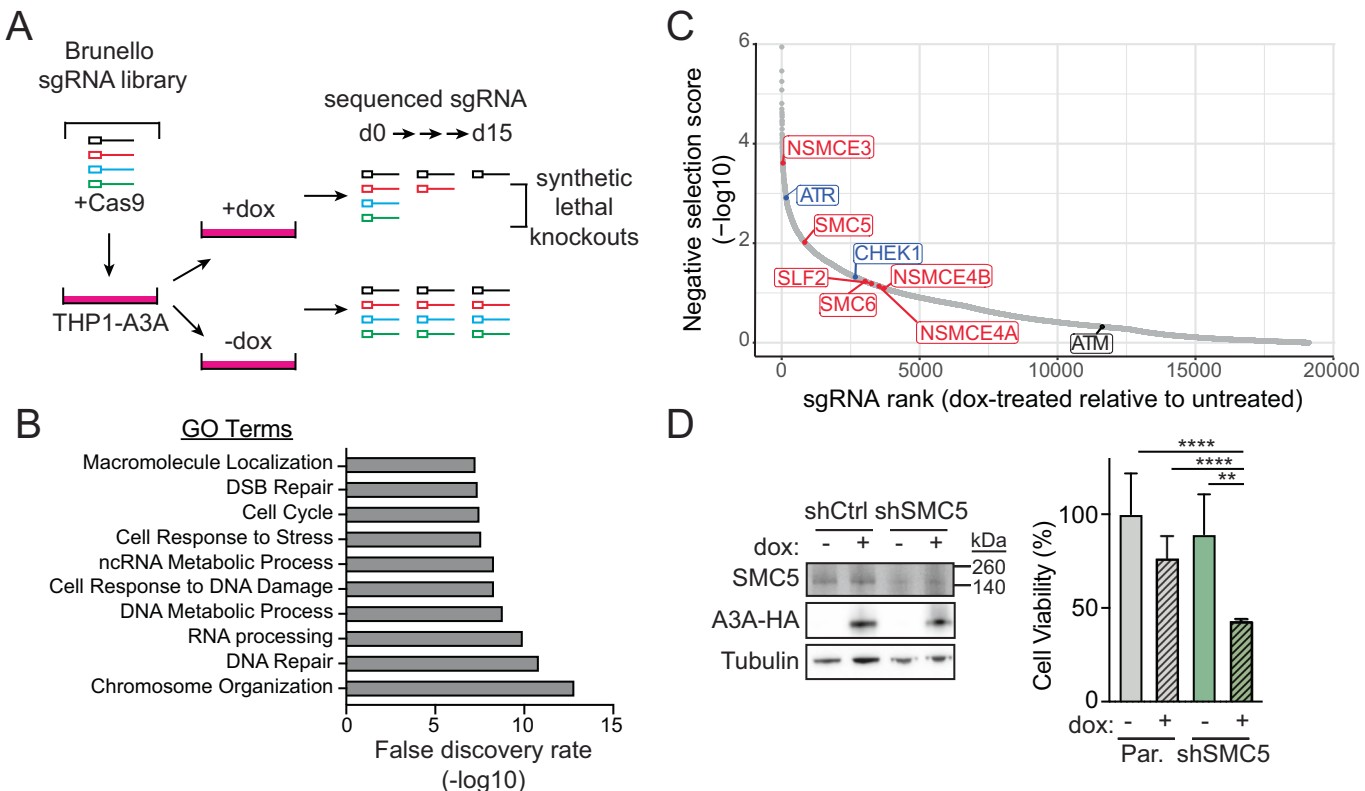

**Figure 1. Functional genomics screen identifies synthetic lethality between loss of the SMC5/6 complex and expression of APOBEC3A.**

(**A**) Schematic for functional genomics screen to identify synthetic lethality with APOBEC3A. The Brunello CRISPR-Cas9 guide RNA (sgRNA) library was used in THP1 cells expressing a doxycycline (dox)-inducible, HA-tagged APOBEC3A transgene (THP1-A3A). sgRNAs were identified and quantified by sequencing at day 0 (baseline library integration) and day 15 after dox treatment. Depletion of sgRNAs at day 15 in dox-treated cells relative to untreated controls represents potential synthetic lethal genes. (**B**) The top 250 genes identified as potentially synthetic lethal with APOBEC3A are grouped by Gene Ontology (GO) terms. (**C**) Negatively selected sgRNAs in dox-treated relative to untreated cells at day 15. SMC5/6 complex genes in red. Previously defined synthetic lethal interactions are denoted in blue. (**D**) THP1-A3A cells were depleted of SMC5 by stable integration of shRNA. Cell lysates were probed with antibodies to HA and SMC5. Tubulin was used as a loading control. The viability of cells treated with dox for 72 h or untreated was determined by FACS after staining for fluorescent-labeled calcein AM (live) and DNA (dead). The mean and SD of triplicate experiments are shown. *p* values by two-tailed *t*-test. ****p < 0.0001 **p < 0.01. Source data are available online for this figure.

depletion of one gene results in SMC5/6 dysfunction (Gallego-Paez et al, 2014; Potts and Yu, 2005; Venegas et al, 2020). Prior studies demonstrated that Smc5/6 is essential in budding and fission yeast (Lehmann, 2005), and complete deletion of SMC5/6 genes is embryonically lethal in mice (Ju et al, 2013), but conditional depletion of SMC5/6 is tolerated in mammalian cells (Atkins et al, 2020; Venegas et al, 2020). Thus, we used partial and inducible depletion of SMC5/6 complex genes. In K562 (myeloid) and Jurkat (T-cell) leukemia cells, shRNA was used to constitutively deplete SMC5 (Fig. 2A; Appendix Fig. S2a). The HCT116 colorectal carcinoma cells were engineered with auxin-inducible degron (mAID) tags on NSMCE4A and SMC6 subunits for inducible depletion of SMC5/6 upon treatment with indole-3-aceticacid (IAA) as previously described (Appendix Fig. S2f) (Natsume et al, 2016; Venegas et al, 2020). K562, Jurkat, and HCT116 cells were engineered to express dox-inducible APOBEC3A transgenes. The doxycycline dose used induced a level of APOBEC3A expression that resulted in minimal DNA damage. However, combined SMC5/6 depletion and APOBEC3A expression significantly impaired proliferation (Fig. 2B; Appendix Fig. S2b,c). Additionally, DNA damage response signaling significantly increased upon

APOBEC3A expression in cells depleted of SMC5 as detected by increased phosphorylation of histone variant H2AX (γH2AX), a response to DNA breaks (Fig. 2C; Appendix Fig. S2d,g). Consistent with these results, we found increased double-stranded DNA breaks (DSBs) by neutral comet assay (Fig. 2D; Appendix Fig. S2e). We hypothesized that substantial DNA damage would cause cell death. Indeed, the culmination of genotoxicity was reflected by decreased viability of cells with concurrent expression of APOBEC3A and depletion of SMC5/6 (Fig. 2E; Appendix Fig. S2h).

APOBEC3A is known to sensitize cells to ATR inhibition (ATRi) and prior reports indicate a role for yeast Smc5/6 in signaling Rad53 (ATR) activation (Khan et al, 2022), therefore we queried whether ATR and SMC5/6 are epistatic in preventing cell death from APOBEC3A-induced genotoxicity. In cells depleted of SMC5/6, we did not find a difference in phosphorylation of Chk1 at ATR-dependent sites (Appendix Fig. S2i), suggesting that SMC5/6 loss does not alter ATR signaling in the cancer cells tested. We found that SMC5 depletion further sensitized APOBEC3A-expressing cells to ATRi (Fig. 2G; Appendix Fig. S2j), indicating a non-epistatic relationship between ATR and SMC5/6 in genome protection during deaminase activity.

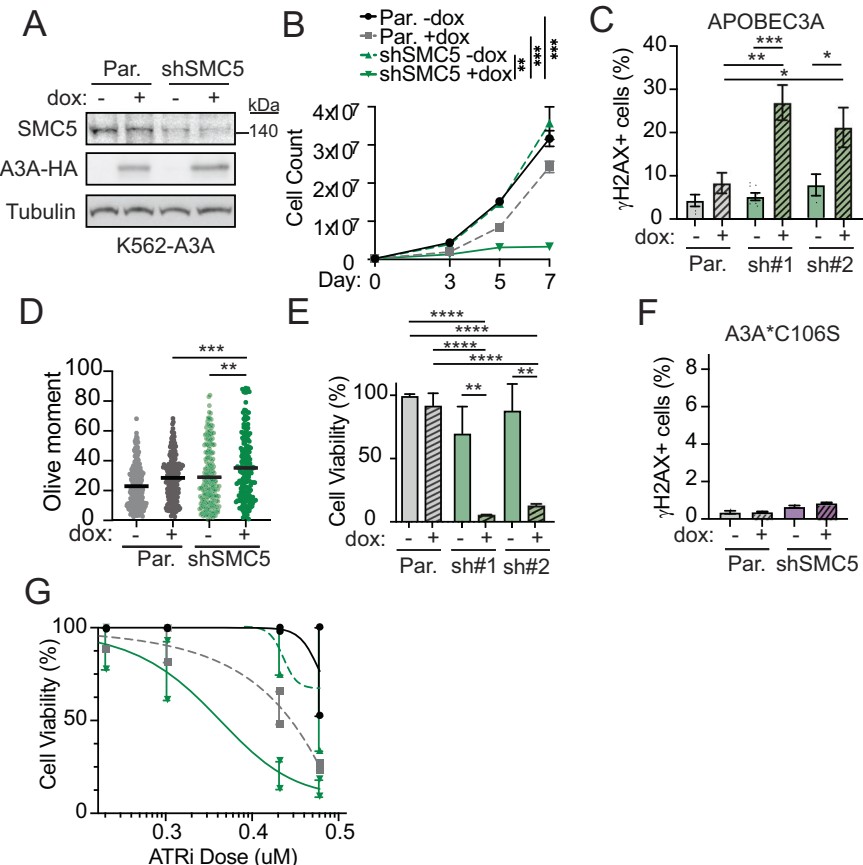

**Figure 2. SMC5/6 loss potentiates APOBEC3A-mediated genotoxicity.**

K562 cells engineered to express doxycycline-inducible HA-tagged APOBEC3A (K562-A3A) were depleted of SMC5 by RNAi (shSMC5) and compared to parental K562-A3A cells. All results are representative of three independent biological replicates. (A) Immunoblot shows APOBEC3A expression (HA antibody) and SMC5 depletion. Tubulin was used as a loading control. (B) Cell proliferation was measured by counting cells over the course of 7 days. Bars are SEM, p value by sum-of-squares F-test. (C) DNA damage response signaling was assessed after 72 h of dox treatment by intracellular staining and flow cytometry analysis of the phosphorylated form of the histone variant H2AX (γH2AX). The mean and SD of triplicate experiments are shown. (D) Comet assay results are shown as a dot plot of individual values, the bar is the median of olive moments. (E) Cell viability was assessed by WST8 live cell quantitation of K562 cells after 7 days of dox treatment. Mean and SD are shown. (F) Intracellular staining and flow cytometry analysis of γH2AX in K562 cells induced with dox for 72 h to express the catalytically inactive A3A*C106S mutant. Mean and SD are shown. (G) The viability of K562-A3A cells was assessed by WST8 quantitation after treatment with ATR inhibitor (AZD6738) for 5 days at indicated doses. Legend as in (B). Data shown as mean and SEM. Data information: for panels (B–F), data were representative of $n = 3$ biological replicates, for panel (G), data were representative of $n = 2$ biological replicates. For panels (C–F), p values by two-tailed t-test. ****$p < 0.0001$, ***$p < 0.001$, **$p < 0.01$, *$p < 0.05$ Source data are available online for this figure.

## APOBEC3A catalytic activity is required for synthetic lethality with SMC5/6 loss

Next, we addressed whether the synthetic lethal phenotype resulting from combined APOBEC3A expression and SMC5/6 loss was due to deaminase-induced genotoxicity. To evaluate the requirement for deamination activity, we constructed K562 cells expressing a catalytically inactive mutant of APOBEC3A containing a C106S amino acid change (Appendix Fig. S3a,b). Cells expressing APOBEC3A-C106S with SMC5 depletion had no differences in proliferation, γH2AX levels, or DSB quantity (Fig. 2F; Appendix Fig. S3c,d). Importantly, SMC5 depletion did not cause increased APOBEC3A deaminase activity (Appendix Fig. S3a). These data demonstrate that deaminase activity is required for the synthetic lethal interaction between APOBEC3A expression and SMC5/6 loss.

Along with APOBEC3A, APOBEC3B has been implicated in tumor mutagenesis (Burns et al, 2013a; Burns et al, 2013b; Caswell et al, 2024; Venkatesan et al, 2021). However, we found that APOBEC3B expression did alter proliferation or H2AX phosphorylation in SMC5-depleted cells (Appendix Fig. S3e–g). Prevention of genotoxicity by SMC5/6 appears to be specific to APOBEC3A among tumor-associated cytosine deaminases.

## SMC5/6 dsDNA binding activity protects cells from APOBEC3A toxicity in yeast

Few studies have addressed the specific functions of SMC5/6 in mammalian cells due to a lack of characterization of mutants that perturb distinct activities of the complex. Recent structural studies of the yeast Smc5/6 have enabled the generation of

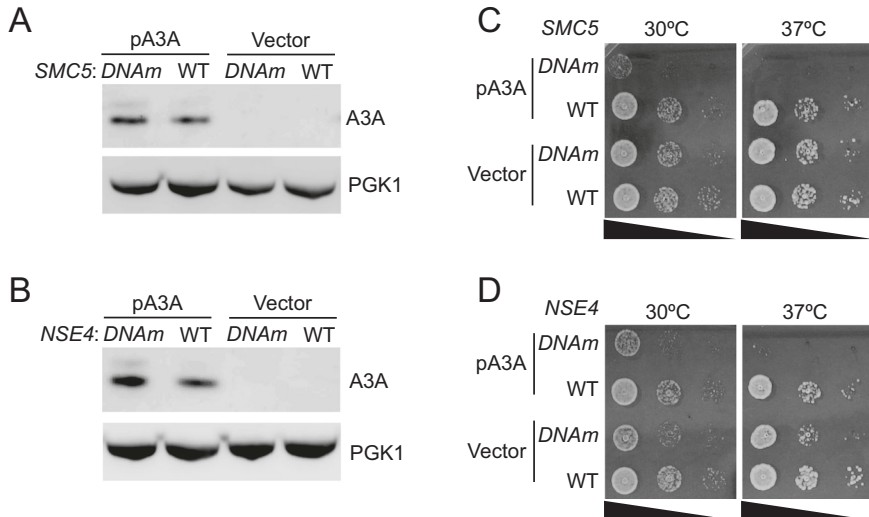

**Figure 3. The DNA binding function of SMC5/6 is essential in yeast cells that express APOBEC3A.**

Wild-type (WT) and mutant yeast cells containing mutations in the DNA binding sites of Smc5 or Nse4 (*DNAm*) were examined. Cells were transformed with APOBEC3A expression plasmid (pA3A) or control vector. (**A, B**) APOBEC3A protein levels from extracts of the indicated cells were assessed by immunoblot using an antibody specific to APOBEC3A. PGK1 was used as a loading control. (**C, D**) Cells with indicated genotypes containing either vector or pA3A were analyzed for growth by spotting tenfold serial dilutions of cells. Plates were grown for 24 h at indicated temperatures. Source data are available online for this figure.

separation-of-function alleles of the complex that impair dsDNA binding activities (Yu et al, 2022). Given that previous studies have established yeast as a model system for studying APOBEC3A activity (Chan et al, 2015; Elango et al, 2019; Hoopes et al, 2016), we asked whether the Smc5/6 protective role against APOBEC3A toxicity is conserved in yeast, and whether its DNA binding activity is required for this protection.

A cryo-EM structure of the dsDNA-bound yeast Smc5/6 complex has identified DNA binding sites on multiple subunits, with several points of contact between each subunit and DNA (Yu et al, 2022). Mutating these sites on the Smc5 and Nse4 subunits led to reduced Smc5/6 chromosome association and extreme sensitivity toward the alkylating agent MMS, suggesting that these mutations may impede DNA replication and repair needed for surviving alkylation damage (Yu et al, 2022). We thus examined whether the dsDNA binding mutant allele of Smc5 (smc5-DNAm, K89, K97, K98, K145, R146, R147, K192 all to A) or Nse4 (nse4-DNAm; R251, R256, R257, R258 all to E) were sensitive to the expression of human APOBEC3A. To do so, we transfected human APOBEC3A under a yeast promotor into mutant or wild-type cells. We confirmed comparable APOBEC3A expression in all cells (Fig. 3A,B). Consistent with prior studies, wild-type cells were tolerant of deaminase activity as they grew similarly to those transfected with empty vector at both 30 and 37 °C (Fig. 3C,D) (Hoopes et al, 2016). In striking contrast, yeast harboring *smc5-DNAm* or *nse4-DNAm* alleles exhibited poor viability upon APOBEC3A expression (Fig. 3C,D). These data demonstrate that SMC5/6 DNA binding is critical for cell growth when APOBEC3A is active in yeast. Given that DNA binding is a fundamental feature of SMC5/6, an extrapolation of this result is that this activity also protects human cells from the genotoxic effects of APOBEC3A.

## APOBEC3A mutagenesis is incompatible with SMC5/6 dysfunction in cancer

To evaluate the interaction of APOBEC3A activity and SMC5/6 loss in human cancers, we quantified APOBEC3A mutational signatures in tumors with deleterious mutations in SMC5/6 subunit genes. Deleterious mutations were defined as exonic missense, nonsense, or frameshift base changes (Choi et al, 2012; McLaren et al, 2016). Within TCGA, 160 tumors with deleterious mutations in SMC5/6 genes were identified (Fig. 4A). SMC5 and SMC6 were the most frequently mutated genes of all subunits (Fig. 4B). For comparison, we defined a control set of tumors with no mutations in SMC5/6 subunit genes (n = 131) that were tissue-matched (Fig. 4A,C). Tumors with dysfunctional SMC5/6 had a higher overall mutation burden (Fig. 4D), consistent with the role of the complex in genome integrity. To determine the source of mutagenesis in tumors with dysfunctional SMC5/6, we examined single base substitution (SBS) signatures. All SBS signatures that comprised more than 4% contribution to mutation burden within each group of tumors are shown (Fig. 4E). The APOBEC3A signatures, SBS2 and SBS13, comprised a substantial portion of mutations in the control tumors but were notably absent in the SMC5/6-mutant tumors (Fig. 4E). These data support our experimental findings that combined dysfunction of SMC5/6 and active APOBEC3A are incompatible in viable human tumors.

## SMC5/6 loss promotes APOBEC3A-mediated DNA damage during replication

It was previously shown that APOBEC3A activity at replication forks results in mutations on both leading and lagging strands, stalled DNA replication, and activation of DNA damage signaling (DeWeerd et al, 2022; Green et al, 2016; Haradhvala et al, 2016; Hoopes et al, 2016; Landry et al, 2011; Seplyarskiy et al, 2016). Damaged replication forks

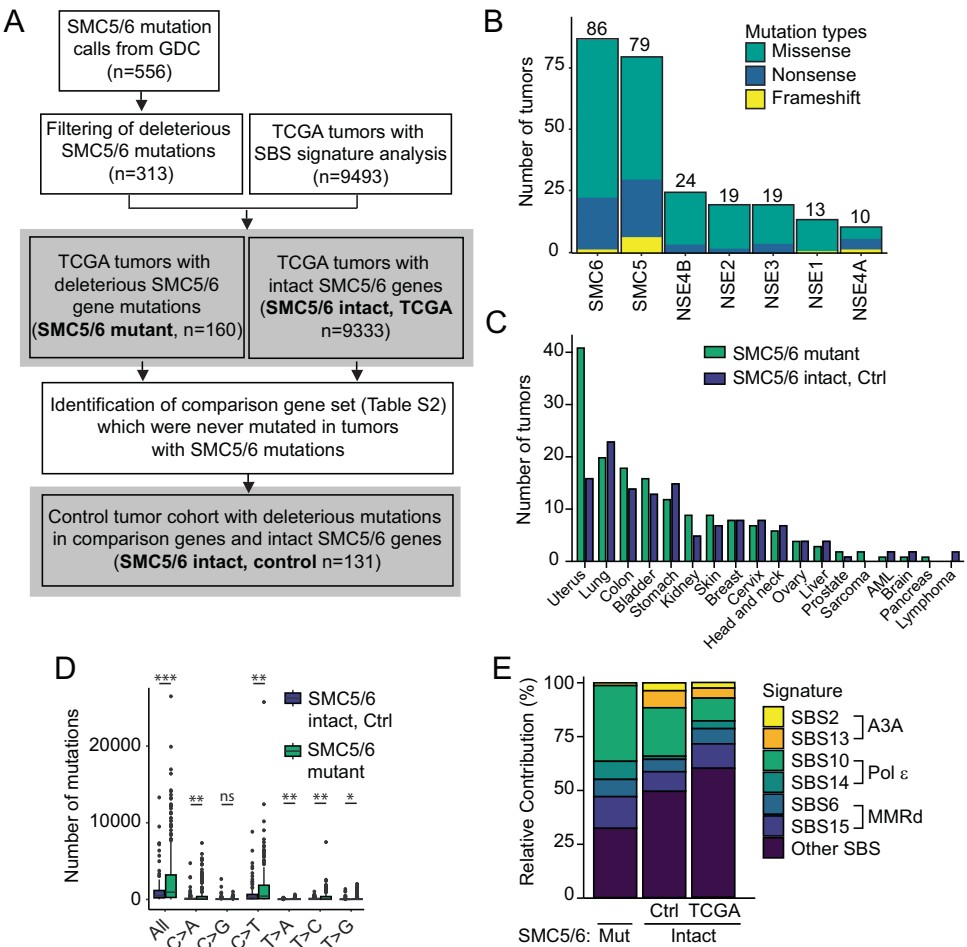

**Figure 4. APOBEC3 mutational signatures are absent in tumors with dysfunctional SMC5/6.**

(A) Pipeline for mutational signature analysis in tumors with SMC5/6 dysfunction. Tumor genomes in the GDC data portal were evaluated for missense, nonsense, or frameshift mutations in the coding regions of SMC5/6 complex genes, which would predict dysfunction. After matching with tumors in TCGA in which SBS signatures were defined, 160 tumors were identified as having mutant SMC5/6 genes. Control genes ($n = 40$, listed in Table EV1) were defined by those which were never mutated in tumors with deleterious mutations in SMC5/6 genes. Tumors in which deleterious mutations in control genes were identified constituted a control (Ctrl) cohort. Bold text in gray squares indicates tumor groups included in panels (B–E). (B) The number of tumors with mutations in each SMC5/6 subunit gene. (C) The type of tumors represented in both SMC5/6 mutant and SMC5/6 intact control groups. (D) Total mutation burden in each tumor genome from SMC5/6 mutant ($n = 160$) and SMC5/6 intact control ($n = 131$) cohorts. The bar is the median. $p$ value by two-tailed $t$-test, $*p < 0.05$, $**p < 0.01$, $***p < 0.001$. (E) The relative contribution of each single base substitution (SBS) mutational signature (COSMIC v3.2) identified within the SMC5/6 mutant (Mut) and SMC5/6 intact cohorts. The latter is divided by the control cohort and all other tumor genomes within TCGA. SBS signatures that comprise >4% relative contribution to mutations within each cohort are included, along with their proposed etiology. Pol ε polymerase epsilon, MMRd mismatch repair deficiency. Source data are available online for this figure.

can result in replication stress and/or DNA breaks. We hypothesized that APOBEC3A activity at replication forks was a source of genotoxicity in SMC5/6-depleted cells. We used immunofluorescent staining of cyclin A to mark replicating K562-A3A cells (Sobczak-Thepot et al, 1993) and γH2AX foci to quantify DNA damage (Fig. 5A; Appendix Fig. S4a). In cells with intact SMC5/6, most APOBEC3A-induced γH2AX foci occurred in replicating cells (Fig. 5B,C; Appendix Fig. S4b). Depletion of SMC5 resulted in increased DNA damage upon APOBEC3A expression, as shown by a significant increase in cells with ≥5 γH2AX foci, nearly all of which occurred in cyclin A-positive cells (Fig. 5B,C; Appendix Fig. S4b). We then used a double thymidine block to synchronize cells at the G1-S junction and followed cells after release for 24 h throughout the cell cycle (Fig. 5D). We observed a significant accumulation of γH2AX in

cells expressing APOBEC3A as they progressed through DNA replication (Fig. 5E). Notably, APOBEC3A-expressing cells depleted of SMC5 accumulated higher levels of γH2AX throughout DNA replication relative to those with intact SMC5 (Fig. 5E). These findings are consistent with prior reports of APOBEC3A causing genome damage during DNA replication (Green et al, 2016; Hoopes et al, 2016; Seplyarskiy et al, 2016), which we now show is exacerbated by the loss of SMC5/6.

Circumstances in which replication forks are stalled may provide more ssDNA substrate for deamination events. SMC5/6 has been implicated in perturbations and control of the G2/M cell cycle checkpoint. In plants, defective SMC5/6 promotes cell cycle progression upon DNA damage despite appropriate activation of the replication checkpoint (Wang et al, 2018). In yeast,

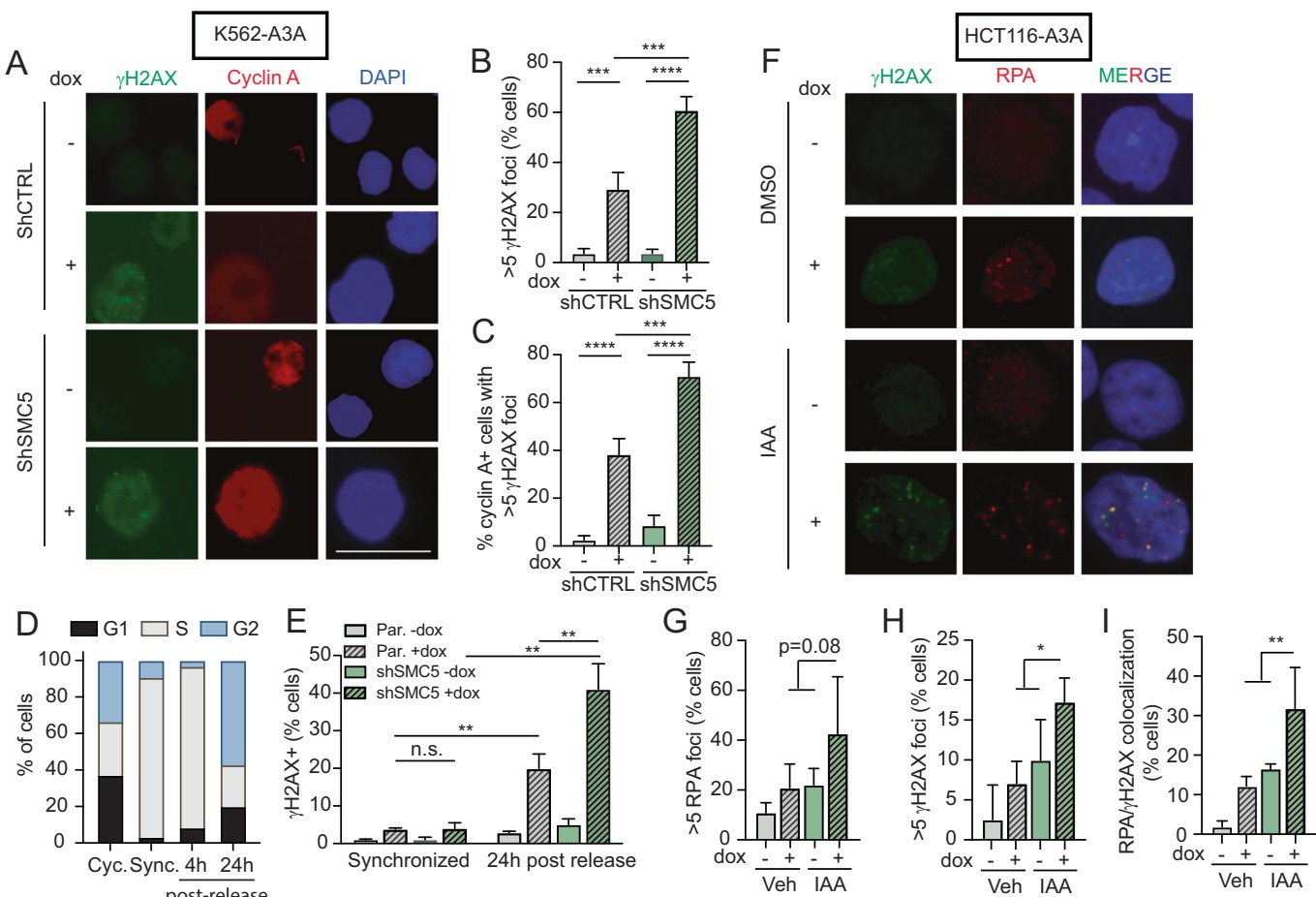

**Figure 5. DNA damage caused by the combination of APOBEC3A activity and SMC5/6 loss occurs during DNA replication.**

(A–C) K562-A3A cells were treated with dox for 72 h and then analyzed by immunofluorescent staining of cyclin A and γH2AX. (A) Representative images are shown. The scale bar is 25 µm. DAPI stains nuclei. (B) Quantification of nuclei with ≥5 γH2AX foci. (C) Quantification of cyclin A-positive cells that had ≥5 nuclear γH2AX foci. (D) K562-A3A cells were synchronized by double thymidine block (sync.) and then released. Dox was added after the first thymidine block. Cells were collected 4 and 24 h after release and compared to asynchronously cycling (cyc.) cells. The cell cycle was analyzed by propidium iodide (PI) staining. Bars are the mean of three biological replicates. (E) Cells were analyzed for intracellular γH2AX staining by flow cytometry after synchronization and release. (F–I) HCT116 cells with integrated E3 OsTIR1 ligase and mAID-tagged NSE4A and SMC6 subunits were treated with IAA to degrade SMC5/6 components and with dox to induce APOBEC3A expression for 72 h. (F) Representative images showing RPA foci, γH2AX foci, and DAPI staining (blue). (G–I) Quantification of cells with >5 RPA foci (G), >5 γH2AX foci (H), and co-localized foci (I). At least 200 nuclei were analyzed per condition. Data information: for panels (B, C, E, H, I) $n = 3$ biological replicates, For panels (B, C, E), $p$ value by two-tailed $t$-test. For panels (G–I) $p$ value by nested Anova ****$p < 0.001$, ***$p < 0.001$, **$p < 0.01$, *$p < 0.05$. Error bars are SEM in panels (B, C) and Error bars are SD in panels (E, G–I). Source data are available online for this figure.

Nse2-mediated SUMOylation of Rqh1, a RecQ helicase, is important for replication checkpoint signaling (Khan et al, 2022). Therefore, we evaluated cell cycle profiles in SMC5/6-depleted cells to determine whether replication fork stalling could explain the exacerbated genotoxicity from APOBEC3A activity. We found that cell cycle profiles were unchanged by SMC5/6 loss (Appendix Fig. S4c). Given the exclusive activity of APOBEC3A on ssDNA substrates (as compared to dsDNA), we evaluated the availability of ssDNA in cells depleted of SMC5/6. We observed that SMC5 depletion did not alter the amount of nascent ssDNA as detected by native BrdU staining (Appendix Fig. S4d). These data show that neither cell cycle perturbations nor ssDNA substrate availability explain the excessive genotoxicity caused by APOBEC3A in the absence of SMC5/6.

We next examined the localization of γH2AX foci with respect to sites of replication stress labeled by RPA foci. Following combined SMC5/6 depletion and APOBEC3A induction in HCT116-A3A cells, we observed an increase in RPA and γH2AX foci relative to controls (Fig. 5F–H). A substantial increase in co-localization of RPA and γH2AX foci was detected in cells with both APOBEC3A expression and SMC5/6 depletion (Fig. 5I). These data demonstrate a physical proximity of replication stress and DSB signaling, which suggests that DNA breaks are arising from damaged replication forks.

## SMC5/6 is required for replication tract lengthening in APOBEC3A-expressing cells

To understand how replication forks were affected by deaminase activity upon SMC5/6 loss, we examined the impact of APOBEC3A activity on replication fork dynamics using single-molecule DNA

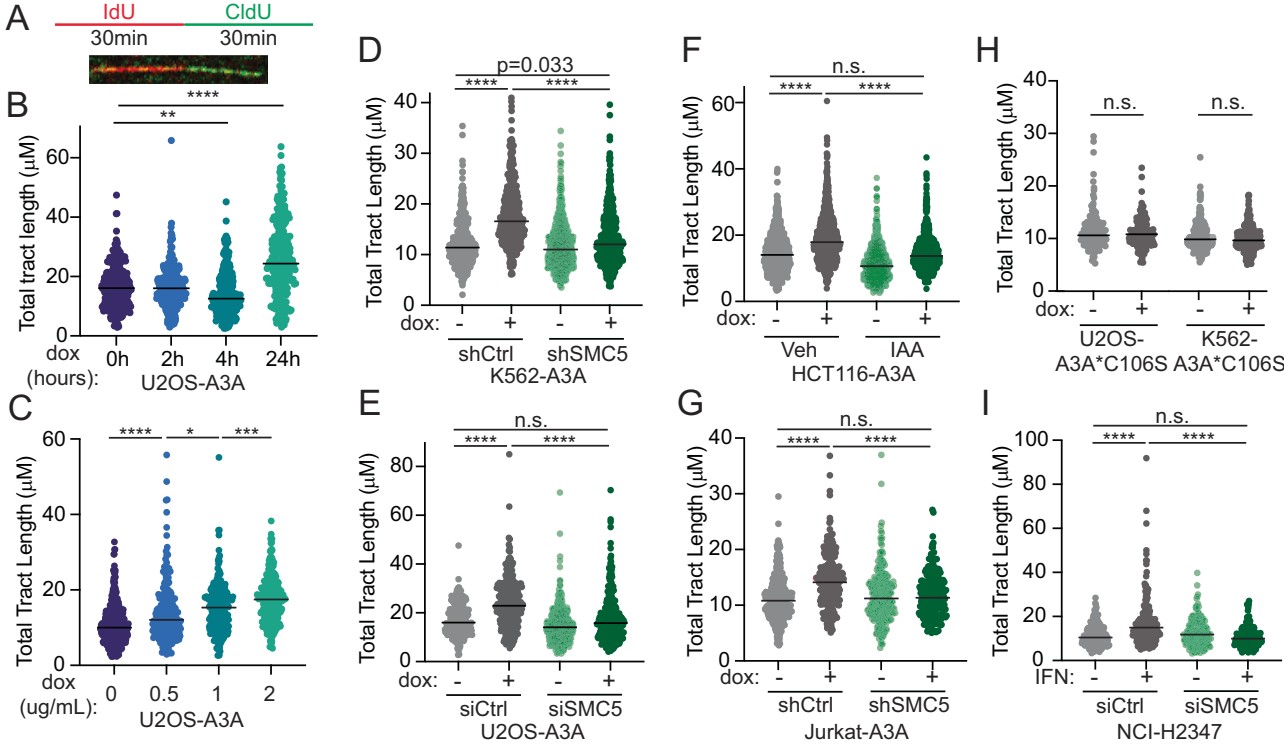

**Figure 6. SMC5/6 is required for APOBEC3A-mediated replication fork elongation.**

(A) Schematic of DNA fiber assay and representative fiber tract. (B, C) Total tract length (μM) of DNA fibers (CldU + IdU) representing complete replication tracts from U2OS-A3A cells treated with 1 ug/ml dox for indicated time points (B) and indicated dox doses for 72 h (C). (D–G) Total tract length (μM) of complete fibers (CldU + IdU) in K562-A3A (D), USOS-A3A (E), HCT116-A3A (F), and Jurkat-A3A cells (G) treated with dox for 72 h. (H) Total tract length (μM) of DNA fibers from K562 and U2OS cells induced with dox for 72 h to express catalytically inactive APOBEC3A (A3A*C106S). (I) Total tract length of DNA fibers from NCI-H2347 cells transfected with siRNA targeting SMC5 or control and treated with type I IFN for 72 h. Data information: for panels (B–I) DNA fiber assays were performed in biological triplicate and analyzed by Kruskal–Wallis test. Bars are median. ****$p < 0.0001$, ***$p < 0.001$, **$p < 0.01$, *$p < 0.05$. Source data are available online for this figure.

fiber spreading (Quinet et al, 2017). Cells were pulsed sequentially with thymidine analogs IdU (red) and CldU (green) for a duration of 30 min each then analyzed for total replication tract length (IdU + CldU) (Fig. 6A). Over a time course of APOBEC3A induction, we found that initial responses to deaminase activity resulted in shorter replication tracts (Fig. 6B) which is consistent with a prior study performed in HCT116 cells (Mehta et al, 2020). Surprisingly, we found that after an extended duration (>24 h), APOBEC3A expression resulted in a dose-dependent increase in total tract length, indicative of replication fork elongation (Fig. 6B,C). APOBEC3A-mediated fork elongation was observed in multiple cell types (Fig. 6D–G) and was dependent on deaminase activity (Fig. 6H). In all cell types, SMC5/6 depletion led to abrogation of APOBEC3A-dependent replication fork lengthening (Fig. 6D–G). Interestingly, SMC5/6 depletion mitigated the fork elongation caused by APOBEC3A yet also resulted in DNA damage and cell death (Fig. 2). In the NCI-H2347 non-small cell lung cancer cell line, we found that type I interferon (IFN) treatment upregulated endogenous APOBEC3A expression (Appendix Fig. S5a) which correlated with an increase in replication tract length (Fig. 6I). A recent study showed that IFN β treatment of cancer cell lines caused accelerated replication speed (Raso et al, 2020). While many effects of IFN may contribute to changes in replication dynamics, we found that SMC5 depletion abrogated

fork elongation in NCI-H2347 cells even in the presence of IFN (Fig. 6I; Appendix S5b). Thus, results from cells expressing endogenous APOBEC3A mimic those from cells with ectopic expression. Together, these data suggest that the activity of SMC5/6 which enables replication elongation in the context of APOBEC3A activity is protective against genotoxicity.

## PrimPol promotes APOBEC3A-mediated elongation of replication tracts

Next, we sought to determine the mechanism by which APOBEC3A leads to longer replication tracts. APOBEC3A catalyzes the conversion of cytidine to uracil, which is excised by DNA glycosylases, leaving an abasic site (Chen et al, 2006; Richardson et al, 2014), which presents an obstacle for replicative polymerases. The dual primase-polymerase, PrimPol, is capable of re-initiating DNA synthesis downstream of replication obstacles (Mouron et al, 2013; Quinet et al, 2021; Quinet et al, 2020). To determine if PrimPol functions in repriming downstream of APOBEC3A-mediated DNA lesions, we expressed APOBEC3A or empty vector (EV) in PrimPol-knockout U2OS cells (Quinet et al, 2020) and measured replication tract length. Loss of PrimPol abrogated the replication tract lengthening generated by expression of APOBEC3A (Fig. 7A).

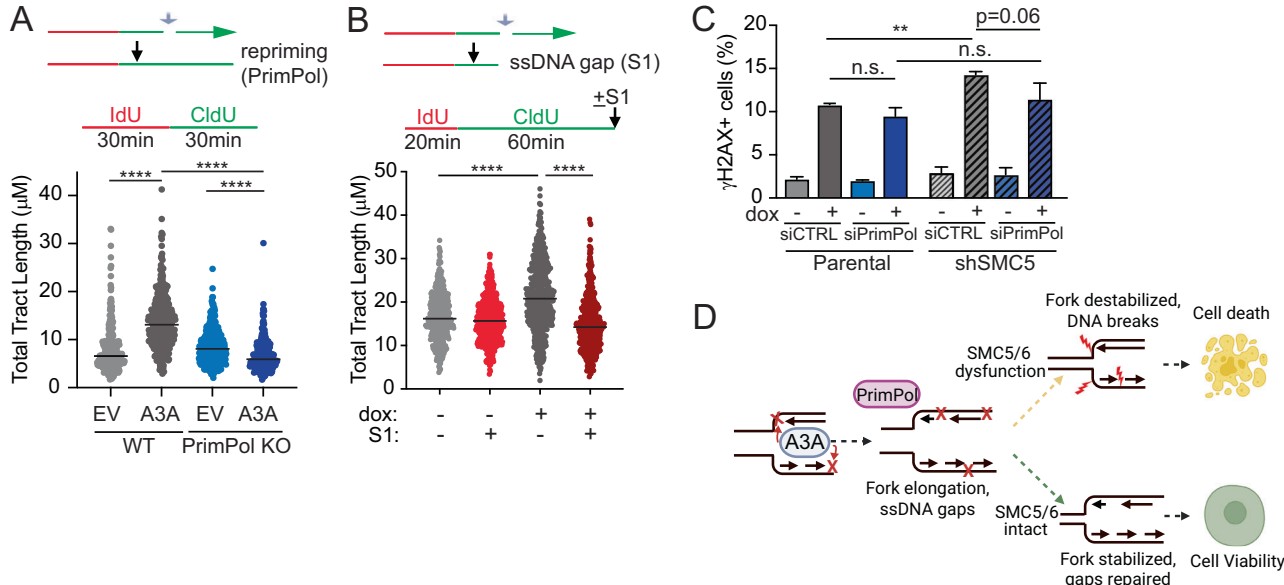

**Figure 7. APOBEC3A-mediated replication fork elongation is dependent on PrimPol.**

(A) Depiction of replication fork lesion bypassed by PrimPol-mediated repriming downstream of the lesion (gray arrow), resulting in longer DNA fiber. Schematic of fiber assay. U2OS cells depleted of PrimPol by CRISPR-Cas9 editing were compared to isogenic controls. Total tract length (CldU+IdU) of DNA fibers was measured 24 h after transfection of empty vector (EV) or APOBEC3A (A3A). (B) Depiction of replication fork lesion with ssDNA gap (gray arrow) which would be expected to result in shorter DNA fiber upon cleavage of the ssDNA gap by S1 nuclease. Schematic of S1 fiber assay. Total tract length (CldU+IdU) of U2OS-A3A cells treated with dox for 72 h. DNA fiber assays in panels (A, B) were performed in biological triplicate and analyzed by Kruskal–Wallis test. Bars are median, ****$p < 0.0001$. (C) K562-A3A cells were depleted of SMC5 by integrated shRNA and/or PrimPol by siRNA transfection. Cells were treated with dox for 72 h then analyzed for γH2AX by flow cytometry. Error bars are SD of $n = 3$ biological replicates. $p$ value by two-tailed $t$-test, **$p < 0.01$. (D) Proposed model of APOBEC3A-induced genotoxicity enabled by SMC5/6 dysfunction. Source data are available online for this figure.

PrimPol-mediated repriming leaves short ssDNA gaps behind the replication fork where damaged DNA was skipped (Fig. 7B) (Quinet et al, 2017; Taglialatela et al, 2021; Tirman et al, 2021). Post-replicative gaps are too small to be visualized at the resolution of DNA fiber imaging (Quinet et al, 2017), therefore replication tracts undergoing PrimPol-mediated repriming should appear longer despite containing ssDNA gaps. To determine whether APOBEC3A activity results in post-replicative ssDNA gap formation, we used a modified version of the DNA fiber protocol in which genomic DNA is treated with an ssDNA-specific S1 endonuclease after pulse labeling with IdU and CldU (Quinet et al, 2017). Shorter DNA fibers result upon S1 treatment if ssDNA gaps are present (Fig. 7B). Treatment with S1 nuclease led to significantly decreased DNA fiber length in cells with active APOBEC3A (Fig. 7B). Our findings suggest that APOBEC3A likely does not cause an increased rate of DNA synthesis but rather causes apparent elongation of replication tracts due to "skipping" of base lesions by PrimPol.

We queried whether PrimPol loss would cause substantial DNA damage in cells that express APOBEC3A, similar to the phenomenon noted with SMC5/6 loss. Instead, we found that cells depleted of PrimPol did not have increased levels of γH2AX when APOBEC3A was expressed (Fig. 7C; Appendix Fig. S6a,b). These data suggest that PrimPol mediates replication tract lengthening but not genome stability upon APOBEC3A-induced DNA lesions. Additionally, we found that simultaneous depletion of both PrimPol and SMC5 slightly decreased the number of cells with γH2AX staining in APOBEC3A-expressing cells relative to those

with selective SMC5 depletion. These results suggest that PrimPol and SMC5 coordinate a response to APOBEC3A-induced DNA lesions at replication forks (Fig. 7D).

## Discussion

Tumor genome sequencing has demonstrated that mutagenesis from APOBEC3A is widespread throughout human cancers (Alexandrov et al, 2020; Petljak and Alexandrov, 2016), however the mechanisms that enable the dysregulated activity of APO-BEC3A in cancer remain elusive. Several genomic determinants that enhance APOBEC3A activity have been elucidated recently, including a preference for acting at TC dinucleotides, stem-loop structures, and ssDNA at replication forks (Buisson et al, 2019; Buisson et al, 2017; Jalili et al, 2020; Langenbucher et al, 2021; Nik-Zainal et al, 2012; Petljak et al, 2019; Seplyarskiy et al, 2016). The mechanisms by which cells respond to APOBEC3A activity in order to maintain genome integrity have also been examined and include the replication checkpoint (Buisson et al, 2017; Green et al, 2017) as well as HMCES, which protects abasic sites in ssDNA (Biayna et al, 2021; Mehta et al, 2020). These prior studies demonstrate that multiple genome-protective responses are required to prevent cytotoxicity from APOBEC3A. We now report a previously unknown, conserved mechanism of genome protection from APOBEC3A activity enacted by the SMC5/6 complex.

In addition to defining a dependence on SMC5/6, we found that cells with APOBEC3A expression exhibit elongated replication

tracts relative to controls. This is counterintuitive to what would be expected of a response to base damage in ssDNA. Prior studies have found that conditions in which replication forks accelerate, such as PARP inhibition, are associated with genome instability and DNA damage (Maya-Mendoza et al, 2018; Merchut-Maya et al, 2019; Zhong et al, 2013). In contrast, the lengthening of nascent strands in DNA fibers after APOBEC3A activity is likely due to extension of DNA synthesis beyond obstacles rather than an increased rate of DNA synthesis. While we found that replication tract elongation occurred after 24–72 h of APOBEC3A expression, we found that shorter exposure to APOBEC3A resulted in decreased tract length (Fig. 6B). These data are consistent with a prior study found that brief APOBEC3A expression within 4 h caused shorter replication tracts (Mehta et al, 2020). We posit that the differences between these two findings reflect early versus late responses to APOBEC3A activity; deamination-induced damage may initially result in fork slowing or stalling, however the same forks may be able to progress upon activation or recruitment of DNA damage tolerance pathways. Similar to our findings, a recent study demonstrated that APOBEC3A activity leads to ssDNA gaps in replication tracts, which is dependent on PrimPol (Kawale et al, 2024). Additionally, in that study, inhibition of fork reversal, a DNA damage tolerance mechanism that results in shorter tracts, resulted in increased ssDNA gaps indicative of a shift to PrimPol activity. These data, along with our findings, suggest that multiple fork protection mechanisms are capable of managing damage caused by deamination, and may have varied effects on tract length. We envision a time-dependent adaptation to APOBEC3A activity in PrimPol upregulation and recruitment to stalled forks with subsequent re-initiation of DNA synthesis. Indeed, in a study of cisplatin-induced replication stress, PrimPol was found to be upregulated and chromatin-associated only upon treatment with a second dose of cisplatin (Quinet et al, 2020). In fact, we found that fork lengthening in response to APOBEC3A was dependent on PrimPol, which may not be recruited to replication forks immediately upon deamination.

The SMC5/6 complex has structural and catalytic functions, all of which have been demonstrated to play roles in genome stability (Aragon, 2018; Peng and Zhao, 2023). Although the function of SMC5/6 in genome maintenance is not fully understood, recent single-molecule studies demonstrate stable binding of Smc5/6 to ssDNA-dsDNA junctions, which mimic DNA replication and repair structures (Chang et al, 2022; Tanasie et al, 2022). This observation suggests that the DNA binding activity of Smc5/6 can be important for protecting junction-containing structures, which increase due to APOBEC3A activity during replication. Our yeast data support this idea. Thus, a potential model to explain the synthetic lethal interaction between APOBEC3A activity and loss of SMC5/6 is that deaminase activity at the replication fork leads to ssDNA gaps generated by PrimPol, which are protected from cleavage by SMC5/6 binding DNA. In this model, SMC5/6 loss can destabilize forks and gaps, leading to DNA breaks and genotoxicity (Fig. 7D). Future studies should determine whether SMC5/6 binding to ssDNA, dsDNA, or ss-dsDNA junctions is key for protecting cells from APOBEC3A-induced lesions. Additionally, the potential for SMC5/6 to prevent genotoxicity from mutagens beyond APOBEC3A is an important future direction.

ATR activity mitigates APOBEC3A-mediated replication fork damage (Buisson et al, 2017; Green et al, 2017). Here we demonstrate that ATR and SMC5/6 act independently to prevent cytotoxicity from APOBEC3A. ATR activity was recently shown to promote repair of APOBEC3A-induced ssDNA gaps (Kawale et al, 2024). Therefore, it is possible that SMC5/6 stabilizes gap-containing structures and ATR acts independently to repair gaps resulting in compounded toxicity when both are inhibited in APOBEC3A-expressing cells. It is alternatively possible that SMC5/6 protects ssDNA gaps is through homology-directed repair (HDR). Post-replicative gaps are repaired in part through template switching, a replication-specific HDR pathway (Tirman et al, 2021). SMC5/6 regulates the resolution of HDR intermediates during replication-associated repair (Ampatzidou et al, 2006; Chen et al, 2009; Irmisch et al, 2009). Additionally, PrimPol has been proposed to prevent mutagenesis from APOBEC3 enzymes by stimulating HDR to limit error-prone DNA synthesis (Pilzecker et al, 2016). Thus, SMC5/6 may be important for a PrimPol-initiated HDR pathway.

A model in which ssDNA gap-containing forks are generated by APOBEC3A activity and require SMC5/6 for stabilization is supported by several of our findings. First, we found that PrimPol loss alone in cells with active APOBEC3A does not cause increased DNA damage. Loss of PrimPol would be predicted to decrease ssDNA gap generation and, therefore, mitigate the potential for the formation of DNA breaks. Additionally, our data demonstrate that PrimPol loss partially rescues DNA damage caused by SMC5/6 loss. In the absence of PrimPol, fewer ssDNA gaps would be expected thus the need for SMC5/6 to prevent cleavage is diminished. Alternative mechanisms of ssDNA gap protection by SMC5/6, such as shielding from nucleases or stabilizing to prevent breaks, would also fit in this model.

A non-epistatic model is also possible in which PrimPol and SMC5/6 may promote fork protection independently of one another. PrimPol may not be the only DNA damage tolerance mechanism that is enabled by SMC5/6. Indeed, we find that yeast, which lacks a PrimPol homolog, can tolerate the expression of APOBEC3A as long as SMC5/6 is intact. In yeast, and perhaps also in mammalian cells, SMC5/6 may enable additional mechanisms of fork protection through fork reversal (Thompson and Cortez, 2020) or gap-filling (Peng and Feng, 2016), as previously proposed. In the future, modeling of functional SMC5/6 mutants in mammalian cells may provide an opportunity to mechanistically examine the consequences of SMC5/6 dysfunction. These studies would also provide specific SMC5/6 activities or subunits that may be targeted for cancer treatment without generating undue toxicity to non-malignant cells.

While SMC5/6 has multiple functions, complex formation relies on all subunits being intact (Gallego-Paez et al, 2014; Potts and Yu, 2005; Venegas et al, 2020). Germline defects in SMC5/6 subunits associated with human diseases provide insight into how compromise of a single SMC5/6 subunit disrupts genome stability. For example, SMC5/6-destabilizing mutations in NSMCE3 cause lung disease-immunodeficiency-chromosomal breakage syndrome (LICS) (van der Crabben et al, 2016), and NSMCE2 or SMC5 mutations result in primordial dwarfism and insulin resistance (Payne et al, 2014; Zhu et al, 2023). In this study, we find that deleterious mutations in SMC5/6 subunits correlate with high tumor mutational burdens in human cancers. Our

findings are consistent with a recent in silico study showing that tumors with alterations in SMC5/6 genes display markers of genome instability, such as aneuploidy (Roy et al, 2023). These data raise questions regarding the etiology of mutagenesis and the sensitivity of those tumors to genotoxic agents. Interestingly, we find that SBS10 and SBS14, signatures consistent with DNA polymerase epsilon (pol ε) dysfunction, are over-represented in SMC5/6-mutant tumors. Experimental data in yeast suggest that SUMOylation of pol ε by SMC5/6 promotes DNA synthesis (Meng et al, 2019; Winczura et al, 2019). Our findings in human cancers indicate a similar dependence of pol ε function on SMC5/6. Identification of additional mutational processes in SMC5/6-mutant tumors may provide insight into the contexts in which SMC5/6 dysfunction is permissive of mutagenesis. These studies may indicate opportunities to exploit mutagenesis in tumors with dysfunctional SMC5/6 as a therapeutic vulnerability.

# Methods

## Human cell culture, small molecules, and plasmid transfection

HCT116-NSE4A/SMC6-mAID, a kind gift from the lab of Ian Hickson (Venegas et al, 2020), U2OS-A3A, U2OS PrimPol knockout (Quinet et al, 2020), and 293T cells (used for lentiviral production) were maintained in DMEM media supplemented with 10% tetracycline-free FBS and 1% Pen-Strep. THP1, K562, Jurkat, and NCI-H2347 cells were maintained in RPMI media supplemented with 10% tetracycline-free FBS and 1% Pen-Strep. Except where indicated, all cell lines were purchased from ATCC and tested for mycoplasma at least twice per year. All cells were grown at 37 °C in a humidified atmosphere containing 5% $CO_2$. Cells were treated with 1000 U/ml of type I interferon (Biotechne) every 48 h. The ATR inhibitor AZD6738 was added to the media at doses and time points indicated. Vehicle (DMSO) was added to controls. Expression vectors containing APOBEC3A (pcDNA-A3A-GFP) (Landry et al, 2011) or GFP alone (pcDNA-GFP) were transfected using Lipofectamine 2000 (Thermo Fisher).

## Lentivectors and cell line generation

THP1-A3A, U2OS-A3A, and U2OS-C106S cells were generated by lentiviral transduction as previously described (Everett et al, 2009; Green et al, 2017; Green et al, 2016; Landry et al, 2011). Cas9 was introduced to THP1-A3A using lentivirus (lenti-Cas9-blast) (Sanjana et al, 2014). Blasticidin (Santa Cruz) selection began 24 h after transduction until non-transduced controls were 100% non-viable. Inducible HCT116-NSE4A/SMC6-mAID-A3A were generated by lentiviral transduction using a dox-inducible pFLRU-A3A lentivector with Thy1.2 selection marker. Cells were bead sorted using magnetic anti-Thy1.2 beads (Miltenyi) until a stable >95% Thy1.2+ population was achieved. K562-A3A, K562-A3A*C106S, and Jurkat-A3A were generated by lentiviral transduction using the dox-inducible pSLIK-A3A lentivector with G418 resistance as previously described (Green et al, 2017). All A3A transgenes have a C-terminal HA tag.

## Genome-wide CRISPR-Cas9 knockout screen in THP1-A3A cells

### Functional screen

Pooled lentivirus encoding the Brunello guide RNA library was generated as previously described (Shalem et al, 2014). Large-scale spinfection was carried out with the same conditions described above, using 12-well plates with $2 \times 10^6$ cells per well. Each well was transduced with 50 µl Brunello library lentivirus. Wells were pooled into 15 cm plates after spinfection and overnight incubation and selected using puromycin for 7 days. Following puromycin selection, THP1-A3A-Cas9 cells were plated in triplicate into a 12-well plate at a concentration of $2 \times 10^6$ cells per well. Doxycycline was added to +dox wells every 48 h beginning on day 0. About $400 \times 10^6$ cells cultured in parallel received vehicle control (water) at equal volume. Cells from dox-treated and untreated wells were harvested on the day of dox induction and after 15 days of dox treatment. Genomic DNA was extracted using a Genomic DNA mini kit (Invitrogen) on a pre-PCR bench under sterile conditions to avoid DNA contamination.

### Amplification and sequencing of library gRNAs

Guide RNAs were amplified by PCR from cellular genomic DNA and amplified using one-step PCR with barcodes on reverse primers, as previously described (Shalem et al, 2014). Illumina next-generation sequencing was applied to an amplicon generated from each integrated gRNA (Shalem et al, 2014). Briefly, we used all collected gDNA (1000× coverage) divided into 100 µL PCR reactions with 5 µg of DNA per reaction. Takara ExTaq DNA Polymerase was used with the following PCR program: [95° 2 min (98° 10 s, 60° 30 s, 72° 30 s) × 24, 72° 5 min]. PCR products were gel-purified using the QiaQuick Gel Extraction Kit (Qiagen). Quality assessment was done by qubit (for concentration), bioAnalyzer (for size distribution), and Kapa Library Quantification (for clusterable molarity). The purified pooled library was then sequenced on a HiSeq4000 with ~5% PhiX added to the sequencing lane.

### Genome-wide screen analysis

To count the number of reads associated with each sgRNA taken from the raw Fastq file, we first extracted the sgRNA targeting sequencing using a regular expression containing the three nucleotides flanking each side of the sgRNA 20 bp target. sgRNA spacer sequences were then aligned to a preindexed Brunello library (Addgene) using the short-read aligner "bowtie" using parameters (-v 0 -m 1). Data analysis was performed using custom R scripts, which are uploaded to github.com/khayer/CRISPRkat and pipeline steps are summarized in Fig. S1c. Dataset EV1 shows the output only for MAGeCK version 0.5.9.5 where the counts were normalized to the 1000 non-targeting control sgRNA contained in the Brunello library.

## Gene depletion by RNAi

### Short-hairpin RNA

Commercially available shRNA lentiviral vectors (Sigma, TRCN0000147948, TRCN0000148162, TRCN0000147348, and TRCN0000147918) were used to construct SMC5-depleted K562-A3A and Jurkat-A3A cell lines. Cells were infected with shRNA lentivirus, selected in 1 mg/mL puromycin until non-transduced

controls were 100% non-viable. Of note, constitutive shRNA-mediated depletion of SMC5 was lost after 3–4 months in culture. Many vials were frozen after each cell line generation to preserve those with maximal gene depletion. Cells were only cultured for use in experiments for <6 weeks. Confirmation of gene depletion was done at least monthly while cells were in culture.

### Small interfering RNA

Pooled siRNA oligonucleotides (25 pmol) targeting SMC5 (Horizon SMARTpool) were transfected into $1 \times 10^6$ cells using the RNAiMAX transfection reagent (Invitrogen) according to the manufacturer's protocol. Gene depletion was confirmed by immunoblot and/or quantitative PCR.

### Antibodies

Commercially available antibodies used for immunoblotting, immunofluorescence, intracellular staining, and DNA fiber spreading were obtained from Santa Cruz Biotechnology (Tubulin, Ku86, SMC5, and SMC6), GeneTex (SMC5), Novus Biologicals (Cas9 Antibody 7A9-3A3), Abcam (RPA and BrdU), Biolegend (HA), Cell Signaling (HA, γH2AX, cyclin A, and pChk1-S317), Invitrogen (PGK1), the NIH AIDS Reagent Program (APOBEC3A/B), and BD Biosciences (γH2AX-488, γH2AX-647, and BrdU). Secondary antibodies for immunoblotting were obtained from Jackson ImmunoResearch (goat anti-rabbit IgG, goat anti-mouse IgG). Secondary antibodies for immunofluorescence were obtained from Invitrogen (Alexa Fluor 488 goat anti-mouse IgG, Alexa Fluor 568 goat anti-rabbit IgG). Secondary antibodies for DNA fiber spreading were obtained from Invitrogen (Alexa Fluor 488 chicken anti-rat IgG, Alexa Fluor 546 goat anti-mouse IgG).

### Viability assays

To assess the proportions of live and dead cells, staining was performed using the Live/Dead Kit (Invitrogen) according to the manufacturer's instructions. Data were collected using a Fortessa Flow Cytometer (BD Biosciences) or Accuri C6 Flow Cytometer (BD Biosciences) and analyzed by FlowJo software. To assess viability by metabolic activity, cells were plated in triplicate in a 96-well plate, precultured for 24 h, and then 50 μl media with and without doxycycline was added to each well every other day. About 10 μl WST8 reagent from the Cell Counting Kit-8 (Dojindo) was added to each well 4–6 h prior to analysis using a microplate reader (BMG Labtech Omega).

### Proliferation assay

On day 0, cells were plated at a density of 200,000 cells per well in a six-well plate. Each cell type was grown in the presence and absence of 1 μg/mL doxycycline. Cells grown in the presence of doxycycline received doxycycline doses every other day. On days 3, 5, and 7, data were collected using an automatic cell counter (Countess, Thermo Fisher).

### Intracellular γH2AX detection by flow cytometry

Cells were harvested, fixed, and permeabilized using reagents from the CytoFix/CytoPerm Kit (BD Biosciences) according to the manufacturer's instructions. Cells were stained with a fluorophore-conjugated γH2AX antibody (BD Alexa Fluor 488 or 647 Mouse Anti-H2AX (pS139)) at a ratio of 10 μl antibody per 100 μl cells ($<1 \times 10^6$ cells/sample). Data were collected using a Fortessa Flow Cytometer (BD Biosciences) or Accuri C6 Flow Cytometer (BD Biosciences) and analyzed by FlowJo software.

### Cell synchronization and cell cycle analysis

Cell synchronization was achieved by double thymidine block as previously described (Chen and Deng, 2018), with the following minor modifications: 2 mM thymidine was added to cells for 24 h then removed by change of media. After 9 h recovery, thymidine was again added for 24 h. Following the removal of the second thymidine pulse, cells were analyzed at time 0 and released into thymidine-free media. To analyze the cell cycle, cells were fixed in 70% ice-cold ethanol, washed in PBS, and resuspended in staining solution containing Triton X, RNAseA, and 1 mg/mL propidium iodide (Biotum). Data were collected using an Accuri C6 Flow Cytometer and analyzed by FlowJo software.

### Immunoblotting and immunofluorescence

Cell lysates were prepared by harvesting cells in LDS buffer and boiling for 15 min, then adding 20% β−mercaptoethanol. Lysates were run on Bis-Tris gels and transferred to a nitrocellulose membrane. After incubation with primary and secondary antibodies, membranes were developed using ECL Western blotting reagents (Pierce) on a GelDoc Go system (BioRad). For immunofluorescence, cells were cultured on coverslips. Following treatment, cells were pre-extracted using 0.5% Triton X in PBS for 15 min on ice to visualize chromatin-bound proteins (i.e., RPA). All other immunofluorescence experiments proceeded as follows: cells were fixed with 4% paraformaldehyde for 15 min at room temperature (RT), permeabilized with 0.5% Triton X for 10 min at RT, and blocked with 5% BSA for 1 h at RT. Primary antibodies were diluted in 5% BSA and incubated with slides for 1 h to overnight. Secondary antibodies used were anti-mouse or rabbit Alexa Fluor 488 and 568 (BD Biosciences). Nuclei were visualized by 4.6-diamidino-2-phenylindole (DAPI, Thermo Fisher). Images were acquired using an inverted fluorescent microscope with an attached camera (Leica) and processed using ImageJ. Protein foci and cell staining were analyzed in a blinded fashion.

### Comet assay

Neutral comet assays were performed using CometAssay (Trevigen) according to the manufacturer's protocol with minor modifications as previously described (Wood et al, 2020). In brief, cells were harvested and resuspended at $3 \times 10^5$ cells/mL in ice-cold PBS, combined with molten LMAgarose, plated onto a comet slide, and allowed to dry at 4 °C. Slides were incubated in lysis solution for 1 h at 4 °C and then immersed in 1X TBE buffer for 30 min at 4 °C. Then slides underwent electrophoresis at 25 V for 30–45 min at 4 °C in 1X TBE buffer. After electrophoresis, slides were washed in DNA precipitation solution (1 M ammonium acetate, 95% ethanol) and fixed in 70% ethanol for 30 min. Fixed slides were dried overnight at room temperature in the dark, stained with 1X SYBR Gold (Applied Biosystems), and washed twice with water.

Images were acquired using a fluorescence microscope (Leica). Images were scored using the OpenComet plugin in ImageJ.

## Yeast strains and genetic techniques

All strains used are in W303 background (*ade2-1 can1-100 ura3-1 his3-11,15, leu2-3, 112 trp1-1 rad5-535*) containing wild-type *RAD5*. *smc5-DNAm* and *nse4-DNAm* strains are from Yu, et al (2022). APOBCE3A expression plasmid containing hygromycin drug-resistant marker is a derivative of pySR419-A3A (Hoopes et al, 2016) (gifted from Dr. Steven A. Roberts). This plasmid and its control vector were transformed into yeast cells individually using the standard method and cells were grown on YPD plates containing hygromycin (300 µg/mL) at 30 °C for 48 h. Cell growth for three independent transformants in each case were assessed at 30 and 37 °C.

## DNA fiber assay

U2OS and HCT116 cells were first pulse-labeled for 30 min with 20 µM IdU, washed three times with 1X DPBS, and then pulsed with 100 µM CldU for 30 min. For K562 and Jurkat cells, cells were first pulsed with 20 µM IdU and then flushed with 100 µM CldU for 30 min. After pulse, cells were harvested and collected in ice-cold DPBS (~1500 cells/µL). For the DNA fiber assay with the ssDNA-specific S1 nuclease (S1 Fiber), cells were permeabilized with CSK100 (100 mM NaCl, 10 mM MOPS pH 7, 3 mM $MgCl_2$, 300 mM sucrose and 0.5% Triton X-100 in water) after the CldU pulse for 10 min at room temperature, treated with the S1 nuclease (Thermo Fisher Scientific) at 20 U/mL in S1 buffer (30 mM sodium acetate pH 4.6, 10 mM zinc acetate, 5% glycerol, 50 mM NaCl in water) for 30 min at 37 °C, and collected in PBS-0.1% BSA with cell scraper. Nuclei were then pelleted at ~4600×*g* for 5 min at 4 °C, then resuspended in PBS (nuclei cannot be quantified, so an initial number of cells plated should be considered when resuspending to a final concentration of 1500 nuclei/µl). To spread fibers, 2 µL of cell solution was placed on a charged glass slide, mixed with 6 µL of lysis buffer (200 mM Tris-HCl pH 7.4, 0.5% SDS, 50 mM EDTA), and gravity was used to spread DNA fibers. DNA fibers were fixed in a 3:1 solution of methanol and acetic acid, denatured in 2.5 M HCl for 1 h, and blocked in pre-warmed 5% BSA at 37 °C for 1 h. IdU and CldU were detected using mouse anti-BrdU (1:20, Invitrogen) and rat anti-BrdU (1:75, Abcam), respectively for 1.5 h at room temperature in a humid chamber followed by anti-mouse Alexa-546 (1:50) and anti-rat Alexa-488 (1:50) for 1 h at room temperature in a humid chamber. Slides were mounted with Prolong Gold Antifade Solution (Invitrogen) and cured overnight at room temperature, protected from light. Fibers were imaged with a 63X oil objective on a Leica DM4 B. Quantification and measurement of fibers was done in ImageJ by blinded analysis.

## Mutational signature analysis

Mutation calls of the SMC5/6 complex genes were obtained from the Genomic Data Commons Data Portal at https://docs.gdc.cancer.gov (v.36) (Grossman et al, 2016). Those predicted to cause negative effects on the proteins' functions by either Ensembl VEP (McLaren et al, 2016) or SIFT (Choi et al, 2012) were classified as deleterious mutations. Additionally, single base substitutions (SBSs) of 9493

TCGA tumors were obtained from the COSMIC database at https://www.synapse.org/#!Synapse:syn11726601/files. Tumors with one or more deleterious mutations in the SMC5/6 complex were defined by merging these two datasets and were later used in downstream analysis. In contrast, other genes mutated in tumors with intact SMC5/6 complex were used as a comparison gene set (Table EV1). A set of control tumors was then defined as those which carried detrimental mutations of the comparison gene set. Differences in mutational burden and APOBEC enrichment between samples were inspected and visualized using the R packages ggplot2 and tidyverse, while the statistical difference was accessed by a two-sided Mann–Whitney test. The R package MutationalPatterns (Manders et al, 2022) was used to study the contribution of other COSMIC SBS signatures (v3.2).

## Statistical analysis

All statistical tests were performed in R or GraphPad (Prism). Biological and/or technical triplicate tests were used to ensure robustness and reproducibility of data. Standard deviations, standard error of the mean, and *p* values were generated using paired and unpaired two-tail *t*-tests, *F*-tests, or Anova.

# Data availability

All data presented in this manuscript are available from the corresponding author upon request. Images from Fig. 5 are available at Bioimage Archive (S-BIAD1152).

The source data of this paper are collected in the following database record: biostudies:S-SCDT-10_1038-S44318-024-00137-x.

# Peer review information

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

## Acknowledgements

We thank Dr. Jieya Shao, Dr. Sebastien Landry, and all members of the Green Lab for helpful discussions and input. We thank Stephen Sykes, Lianna Valin, Ian Hickson, Luis Batista, and members of the Vindigni and Bednarski Labs for sharing reagents and technical support. Special thanks to Dr. Steven Roberts for providing expression plasmids and advice regarding APOBEC3A expression in yeast. This study was funded by support from the NIH T32 HL007088 (DFF), NIH K08 CA212299 (AMG), DOD CA200867 (AMG), NIGMS R35 GM145260 (XZ), and NIH Cancer Center Support Grant P30 CA008748 (MSKCC), Cancer Research Foundation, Children's Discovery Institute, and American Cancer Society. Work in the AV lab was supported by the National Cancer Institute (NCI) grants R01CA237263 and R01CA248526.

## Author contributions

**Dylan F Fingerman**: Conceptualization; Data curation; Formal analysis; Investigation; Writing—review and editing. **David R O'Leary**: Conceptualization; Investigation. **Ava R Hansen**: Conceptualization; Investigation; Writing—original draft. **Thi A Tran**: Data curation; Formal analysis; Methodology. **Brooke R Harris**: Data curation; Investigation; Visualization. **Rachel A DeWeerd**: Investigation; Visualization; Methodology; Writing—review and editing. **Katharina E Hayer**: Formal analysis; Validation; Methodology. **Jiayi Fan**: Investigation; Visualization. **Emily Chen**: Investigation. **Mithila Tennakoon**: Investigation. **Alice Meroni**: Methodology. **Julia H Szeto**: Investigation. **Jessica Devenport**: Investigation. **Danielle LaVigne**: Investigation. **Matthew D Weitzman**: Conceptualization; Data curation; Project administration; Writing—review and editing. **Ophir Shalem**: Methodology. **Jeffrey Bednarski** Conceptualization; Writing—review and editing. **Alessandro Vindigni**: Conceptualization; Funding acquisition; Methodology; Project administration. **Xiaolan Zhao**: Conceptualization; Formal analysis; Supervision; Investigation; Writing—original draft. **Abby M Green**: Conceptualization; Data curation; Formal analysis; Supervision; Funding acquisition; Investigation; Visualization; Methodology; Writing—original draft; Project administration; Writing—review and editing.

Source data underlying figure panels in this paper may have individual authorship assigned. Where available, figure panel/source data authorship is listed in the following database record: biostudies:S-SCDT-10_1038-S44318-024-00137-x.

## Disclosure and competing interests statement

The authors declare no competing interests.

