## [Peer Review File · The EMBO Journal]

The SMC5/6 complex prevents genotoxicity upon APOBEC3A-mediated replication stress

Dylan Fingerman, David O'Leary, Ava Hansen, Thi Tran, Brooke Harris, Rachel DeWeerd, Katharina Hayer, Jiayi Fan, Emily Chen, Mithila Tennakoon, Alice Meroni, Julia Szeto, Jessica Devenport, Danielle LaVigne, Matthew Weitzman, Ophir Shalem, Jeffrey Bednarski, Alessandro Vindigni, Xiaolan Zhao, and Abby Green

Corresponding author(s): Abby Green (abby.green@wustl.edu)

Review Timeline:

Submission Date:	22nd Jan 24
Editorial Decision:	9th Feb 24
Revision Received:	21st Apr 24
Editorial Decision:	8th May 24
Revision Received:	9th May 24
Accepted:	17th May 24

Editor: Hartmut Vodermaier

Transaction Report:

Dr. Abby M Green
Washington University
Pediatrics
425 S. Euclid Ave.
MPRB 5105
St. Louis, MO 63110

9th Feb 2024

Re: EMBOJ-2024-116747
The SMC5/6 complex prevents genotoxicity upon APOBEC3A-mediated replication stress

Dear Dr. Green,

Thank you for submitting your study on genetic/functional interactions between SMC5/6 and APOBEC3A to The EMBO Journal. It has now been assessed by three expert referees, whose reports you will find copied below. All referees are in principle supportive of publication, but also raise a number of important issues that would need to be clarified prior to publication. Should you be able to satisfactorily address their comments, we would be interested in considering a revised manuscript further for The EMBO Journal.

I should remind you that it is our policy to allow only a single round of (major) revision, making it important to carefully respond to all points raised at this stage; therefore, please do not hesitate to contact me already during the early stages of your revision work, in case you would like to discuss any of the issues raised by the reviewers, or if you should require an extension of the revision period.

Detailed information on preparing, formatting and uploading a revised manuscript can be found below and in our Guide to Authors. Thank you again for the opportunity to consider this work for The EMBO Journal, and I look forward to your revision in due time.

Yours sincerely,

Hartmut Vodermaier

4) Each main and each Expanded View (EV) figure should be uploaded as individual production-quality files (preferably in .eps, .tif, .jpg formats). For suggestions on figure preparation/layout, please refer to our Figure Preparation Guidelines:

- 5) Point-by-point response letters should include the original referee comments in full together with your detailed responses to them (and to specific editor requests if applicable), and also be uploaded as editable (e.g., .docx) text files.
- 6) Please complete our Author Checklist, and make sure that information entered into the checklist is also reflected in the manuscript; the checklist will be available to readers as part of the Review Process File. A download link is found at the top of our Guide to Authors: embopress.org/page/journal/14602075/authorguide
- 7) All authors listed as (co-)corresponding need to deposit, in their respective author profiles in our submission system, a unique ORCID identifier linked to their name. Please see our Guide to Authors for detailed instructions.
- 8) Please note that supplementary information at EMBO Press has been superseded by the 'Expanded View' for inclusion of additional figures, tables, movies or datasets; with up to five EV Figures being typeset and directly accessible in the HTML version of the article. For details and guidance, please refer to: embopress.org/page/journal/14602075/authorguide#expandedview
- 9) Digital image enhancement is acceptable practice, as long as it accurately represents the original data and conforms to community standards. If a figure has been subjected to significant electronic manipulation, this must be clearly noted in the figure legend and/or the 'Materials and Methods' section. The editors reserve the right to request original versions of figures and the original images that were used to assemble the figure. Finally, we generally encourage uploading of numerical as well as gel/blot image source data; for details see: embopress.org/page/journal/14602075/authorguide#sourcedata

At EMBO Press, we ask authors to provide source data for the main manuscript figures. Our source data coordinator will contact you to discuss which figure panels we would need source data for and will also provide you with helpful tips on how to upload and organize the files.

In the interest of ensuring the conceptual advance provided by the work, we recommend submitting a revision within 3 months (9th May 2024). Please discuss the revision progress ahead of this time with the editor if you require more time to complete the revisions. Use the link below to submit your revision:

Link Not Available

Referee #1:

A3A expression is known to cause genotoxic stress and mutagenesis across many cancer types and it generates a characteristic mutational signature. O'Leary et al. performed genome wide CRISPR screening to identify synthetic lethal interactions with APOBEC3A (A3A) expression in an AML derived cell line THP1-ASA with a constitutive Cas9 transgene. They identified 250 negatively selected genes including ATR and CHEK1 that were previously reported. Chromosome organization was the most significant GO term, and among them multiple components of the SMC5/6 complex were present (SMC5, SMC6, NSMCE3). Using inducible shRNA or an auxin-induced degron in K562, Jurkat and HCT116 cells, they then validated the synthetic lethal interaction between SMC5/6 and A3A expression and activity. Following this they showed that the DNA binding activity of SMC5 and NSE4 is essential for synthetic lethal interaction using A3A expression in yeast. They found that following SMC5 depletion, deaminase-induced DNA damage is maximal during DNA replication and using DNA fiber imaging, they found that A3A activity resulted in longer replication fork tracts that were dependent on SMC5. Using S1 nuclease, they found that this was likely due to PrimPol mediated skipping of A3A induced lesions, resulting in gaps behind the fork. They also provide a compelling correlate in the analysis of human genomes where SMC5/6 mutations appear to be mutually exclusive with A3A mutational signatures in the datasets used for analysis.

Overall, the manuscript provides interesting insights into the mechanism of synthetic lethality between APOBEC3A overexpression and loss of SMC5/6 complex. As A3A expression can be identified in cancers, there is significant interest in both how to exploit this for therapeutic gain, as well as how cells tolerate its expression and its biological effects. The manuscript is consistent with recent findings that A3A generates gaps via PrimPol (Kawale et al. *Sci. Advances* 2024) and they extend these results to demonstrate a novel role for SMC5/6 in the process. Further the results establish a synthetic lethal relationship between A3A and SMC5/6 that is potentially reflected in the relationship of cancer mutational signatures and SMC5/6 inactivation in cancer. The data overall is of high quality and clearly presented. I have a few comments and clarifications that

could be addressed to improve the clarity and impact of the findings.

Major Comments

- 1) The authors demonstrate that A3A over expression resulted in a dose-dependent increase in total tract length that they propose is a result of PrimPol skipping of A3A lesions in Fig 6a. Recently, published work analyzing fork length in 3 different cell lines, including U2OS that was used in this paper, expressing A3A did not observe any increase in tract length (Kawale et al. *Sci. Advances* 2024), although they did implicate PrimPol in lesion skipping and gap formation. The authors should address potential differences between these observations. Are differences in the experimental setup, Dox levels, CldU/IdU timing etc potentially generating different outcomes? Are there other explanations for the discrepancies?
- 2) All of the experiments make use of an inducible system that expresses super-physiological levels of A3A in the cells. Have the authors looked at whether toggling A3A levels in a cell line that expresses it at endogenous levels causes similar effects regarding the synthetic lethality or effects on replication forks? In data from DepMap in AML cells, there is no apparent correlation between SMC5/6 loss and A3A expression levels, however only a couple lines express A3A and this is best addressed within the same cell line background, as there may be many other modulators of the interaction with SMC5/6 loss.
- 3) Nearly all of the validation experiments in the paper make use of a single shRNA for SMC5 from what I can gather. Have any of the key findings been complemented or replicated with an additional shRNA?
- 4) Given the role of SMC5/6 in promoting ATR signaling, is SMC5/6 depletion epistatic with ATR inhibition in the context of A3A expression? In other words, does ATR inhibition bypass SMC5/6 activity?
- 5) In Figure 5 they look at γ H2AX foci in all cells and cyclin A+ cells. There seems to be an increase in both populations (Fig S4b) not just cyclin A+ cells. Can the authors comment on this and how it would fit into their model. The γ H2AX foci are also very difficult to see in their representative images.

Minor comments

- 1) The authors propose multiple times that the SMC5/6-A3A interaction complex could be used as a therapeutic vulnerability in cells expressing A3A or in cells with SMC5/6 mutations. Most cell lines tested appear to require intact SMC5/6 for DNA replication and DNA repair (Payne et al. 2014; van der Crabben et al. 2016; Venegas et al. 2020; Grange et al. 2022; Zhu et al. 2023; DepMap). It would be helpful to elaborate more on how this could be put to use in a therapeutic context.
- 2) I am not sure the journal policy on this point, but none of the western blots have any molecular weight markers, which I suggest correcting.
- 3) In Figures 1, 2 and S2 is there a specific reason for estimating cell viability using three different staining protocols Calcein AM, WST8 and CFSE respectively across THP1, K562 and Jurkat (all suspension) cell lines?
- 4) In Supplementary figure S2b with the western blot of SMC5/6 degradation in HCT116 the A3A is HA tagged and is blotted for the HA tag. Why are there two bands for the HA tag?
- 5) Previous reports like Venegas et al 2020, Cell reports demonstrated induction in DNA damage (γ H2AX) signal upon loss of SMC5/6. In this manuscript, the authors do not see any clear increase in γ H2AX after SMC5 silencing, particularly in the leukemia cell lines (THP1 and K562). Similarly, A3A expression by itself does not appear to induce γ H2AX or the olive movement significantly as shown previously (Wörmann et al., 2021 *Nature cancer*). It might be relevant for authors to discuss this, is this a cell line specific effect? Due to timing of depletion or A3A induction?
- 6) In the results the section titled- "A3A catalytic activity is required for synthetic lethality with SMC5/6 loss", authors mention- "Cells expressing APOBEC3A with SMC5/6 depletion had no differences in proliferation, γ H2AX levels, or DSB quantity (Fig2F and Fig S3c-d)." While I presume the results would be similar, they have only tested the effects of SMC5 silencing and not SMC6.
- 7) The authors mention- "Importantly, SMC5/6 depletion did not cause increased APOBEC3A deaminase activity (Fig S3a)." This data appears to show ~50% reduced A3A activity upon SMC5 silencing. If this reduction is statistically significant and reproducible it might be interesting to discuss the potential molecular pathway at play causing SMC5 silencing to reduce A3A activity.
- 8) The authors tested the requirement of DNA binding activity of the SMC5/6 complex for its synthetic lethal relationship with A3A in yeast. They state "These data demonstrate that a fundamental feature of SMC5/6 in binding dsDNA that is conserved from yeast to human is responsible for supporting cell viability when APOBEC3A is active." As stated, this implies that this is true in both organisms but the requirement of DNA binding for the synthetic interaction has not been directly tested in human cells.
- 9) Authors discuss that SMC5/6 may protect ssDNA gaps through HDR. Are HDR genes enriched in the screen to support this idea? Have the authors performed HDR assays, like the traffic like reporter assay to show that SMC5/6 affect HDR or has this been reported elsewhere?
- 10) The authors report a novel CRISPR screen analysis pipeline. If there is specific code used that goes beyond what is published for the individual tools, it should be provided. It is unclear how these tools are being integrated from the explanation.

Referee #2:

In this manuscript, O'Leary et al. identified the SMC5/6 complex as essential for protecting cells expressing active APOBEC3A (A3A). The authors elegantly demonstrated that cells become more sensitive to A3A-induced DNA damage when depleted of SMC5. Moreover, cells lacking SMC5 and expressing A3A exhibited an increase in DNA damage during replication. Interestingly, A3A overexpression led to an acceleration in replication speed dependent on both SMC5 and PrimPol. From this

study, the authors propose that SMC5/6 represents a potential therapeutic vulnerability in tumors with active A3A.

The manuscript is very well written, and the experiments are well executed with appropriate controls, including the use of multiple cell lines and catalytic mutants. However, some additional experiments are needed to demonstrate the specificity of the synthetic lethal interaction between A3A activity and loss of SMC5/6. Indeed, it is unclear to the reviewer whether the increase in cell death and DNA damage caused by the loss of SMC5/6 occurs only in A3A-expressing cells or also in the presence of different types of DNA damage. Data showing the specificity to A3A of this synthetic lethal interaction would increase the impact of the result presented in this manuscript.

Major comments:

A previous publication reported that A3A overexpression in cells does not slow down the replication fork (PMID: 38241374). Another paper showed that A3A-induced abasic sites slow down replication elongation (PMID: 32492421). How do the authors reconcile their findings (increase in replication track lengthening following A3A expression) with these previous studies showing opposite and conflicting results?

In the discussion, it was speculated that the role of PrimPol in A3A-expressing cells is adaptive over time. Do the authors provide evidence supporting this model? It would be beneficial in reconciling the discrepancy with another study (PMID: 32492421) that shows A3A expression slows down replication in the short term.

Does the increase in cell death, DNA damage levels, and replication speed after A3A expression and depletion of SMC5 depend on the formation of abasic sites caused by UNG2?

It is unclear how the increase in cell death and DNA damage in cells depleted for SMC5 is specific to DNA damage caused by A3A. Does overexpression of A3B lead to similar phenotypes?

Furthermore, does SMC5/6 protect cells against various types of DNA damage that impact replication forks, particularly those involving PrimPol? Or is this protective function specific to DNA damage resulting from cytosine deamination?

Minor comments:

In Figure S2b, there are two bands for SMC6. Is the bottom band corresponding to an SMC6 allele untagged with mAID?

Bar graphs should show individual data points as dot plots.

Can the authors comment on why they did a 1:1 ratio of lentivirus:cell for their CRISPR screen? In pooled CRISPR screens, cells are usually transduced at a relatively low MOI, often between 0.3 and 0.5 to ensure that only a few cells receive more than one gRNA simultaneously.

Suggestions to improve the study:

A previous study by the authors showed that cells expressing A3A are more sensitive to ATRi. Are ATR inhibition and SMC5/6 depletion epistatic and non-epistatic? It would be interesting to determine whether A3A-expressing cells depleted of SMC5/6 show increased sensitivity to ATRi, or if this depletion leads to no further sensitivity.

Referee #3:

In the present study by O'Leary et al, the authors conducted a genetic screen to identify cellular factors exhibiting synthetic lethality with the mutagenic activity caused by APOBEC3A expression. As expected, they found that cells expressing APOBEC3A exhibited synthetic lethality with genes involved in DNA damage repair and chromosome organization. In particular, they found that several subunits of the SMC complex, SMC5/6, appeared on their screen. They subsequently confirmed that depletion of SMC5/6 in cells expressing APOBEC3A causes decreased viability, highlighting the synthetic lethal interaction. Analysis of tumor data from, further supported the genetic interaction as tumors with mutations in SMC5/6 genes lacked APOBEC3A-induced mutational signatures, indicating an incompatibility between SMC5/6 dysfunction and APOBEC3A activity in viable tumors.

Further investigation revealed that APOBEC3A-induced DNA damage occurred primarily during DNA replication and caused high levels of damage exacerbated by SMC5/6 loss. The authors used DNA fiber spreading to analyse replication tract lengths and found that APOBEC3A expression causes elongated replication tracts, dependent on the DNA primase-polymerase PrimPol and SMC5/6.

Overall, this is a well-executed study with conclusions that are well supported by the data and it highlights the importance of SMC5/6 in protecting cells from APOBEC3A-induced genotoxicity. Moreover, the replication experiments provide some glimpses of the potential mechanisms likely related to replication fork stabilisation in response to lesions.

Specific comments

The main criticism relates to the yeast experiments and the implication that Smc5/6's role in repairing APOBEC3A lesions relies on its ability to bind dsDNA. I think the authors should be cautious because the complex is known to bind both ssDNA and dsDNA and presently molecular mechanisms for fork stabilisation/repair are unclear, and APOBEC3A-induced lesion occur on ssDNA. The authors should discuss this.

Fingerman, et al

We appreciate the reviewers time spent assessing our manuscript and the overall positive feedback. We have addressed all questions and critiques which has resulted in an improved manuscript, all of which is detailed below. Please note, in the process of revisions the first author listed has changed to Dylan Fingerman.

Referee #1:

A3A expression is known to cause genotoxic stress and mutagenesis across many cancer types and it generates a characteristic mutational signature. O'Leary et al. performed genome wide CRISPR screening to identify synthetic lethal interactions with APOBEC3A (A3A) expression in an AML derived cell line THP1-ASA with a constitutive Cas9 transgene. They identified 250 negatively selected genes including ATR and CHEK1 that were previously reported. Chromosome organization was the most significant GO term, and among them multiple components of the SMC5/6 complex were present (SMC5, SMC6, NSMCE3). Using inducible shRNA or an auxin-induced degron in K562, Jurkat and HCT116 cells, they then validated the syn lethal interaction between SMC5/6 and A3A expression and activity. Following this they showed that the DNA binding activity of SMC5 and NSE4 is essential for synthetic lethal interaction using A3A expression in yeast. They found that following SMC5 depletion, deaminase-induced DNA damage is maximal during DNA replication and using DNA fiber imaging, they found that A3A activity resulted in longer replication fork tracts that were dependent on SMC5. Using S1 nuclease, they found that this was likely due to PrimPol mediated skipping of A3A induced lesions, resulting in gaps behind the fork. They also provide a compelling correlate in the analysis of human genomes where SMC5/6 mutations appear to be mutually exclusive with A3A mutational signatures in the datasets used for analysis.

Overall, the manuscript provides interesting insights into the mechanism of synthetic lethality between APOBEC3A overexpression and loss of SMC5/6 complex. As A3A expression can be identified in cancers, there is significant interest in both how to exploit this for therapeutic gain, as well as how cells tolerate its expression and its biological effects. The manuscript is consistent with recent findings that A3A generates gaps via PrimPol (Kawale et al Sci. Advances 2024) and they extend these results to demonstrate a novel role for SMC5/6 in the process. Further the results establish a synthetic lethal relationship between A3A and SMC5/6 that is potentially reflected in the relationship of cancer mutational signatures and SMC5/6 inactivation in cancer. The data overall is of high quality and clearly presented. I have a few comments and clarifications that could be addressed to improve the clarity and impact of the findings.

Major Comments

1) The authors demonstrate that A3A over expression resulted in a dose-dependent increase in total tract length that they propose is a result of PrimPol skipping of A3A lesions in Fig 6a. Recently, published work analyzing fork length in 3 different cell lines, including U2OS that was used in this paper, expressing A3A did not observe any increase in tract length (Kawale et al Sci. Advances 2024), although they did implicate PrimPol in lesion skipping and gap formation. The authors should address potential

differences between these observations. Are differences in the experimental setup, Dox levels, CldU/IdU timing etc potentially generating different outcomes? Are there other explanations for the discrepancies?

The reviewer raises a very interesting point and one that, after substantial effort, we cannot entirely resolve. We started by considering the technical differences between our study and Kawale et al. They are: 1) Kawale used a 20/60min analogue pulse whereas we used 30/30min, 2) Kawale used 0.2ug/ml dox whereas we used 1ug/ml (in U2OS cells), and 3) Kawale analyzed only IdU tracts whereas we analyzed IdU+CldU tracts. To address #3, we reanalyzed our original data from multiple cell lines by measuring only the second analogue (note that our analogue order is reversed from Kawale et al, therefore the second analogue measured is CldU) but still found a significant elongation of tracts in A3A-expressing cells. To address the differences in pulse durations (#1) and dox doses (#2), we repeated the experiment using the Kawale et al protocol in our cells. Again, we found extension of fiber tracts when APOBEC3A was expressed, even at levels induced by 0.2ug/ml dox.

Similar to data shown in our original Fig. 6b, we found a dose-dependent increase in tract length when APOBEC3A was expressed.

We looked carefully at the Kawale et al paper and, while no statistically significant tract length differences are evident, there is a slight increase in tract length in HCT15 and MCF10A cells upon APOBEC3A induction (Kawale Fig 1d, e). While our new experiments do not explain the differences between our findings and Kawale et al, we think that the differences in cell engineering probably explain the variation. In other words, our cell lines express higher levels of APOBEC3A (even at the same dose of dox, possibly due to increased number of lentiviral integrations or sensitivity of dox promoter), therefore more deaminase activity leads to tract length increases. We have included the Kawale, et al paper in our revised discussion on page 18.

2) All of the experiments make use of an inducible system that expresses super-physiological levels of A3A in the cells. Have the authors looked at whether toggling A3A levels in a cell line that expresses it at endogenous levels causes similar effects regarding the synthetic lethality or effects on replication forks? In data from DepMap in AML cells, there is no apparent correlation between SMC5/6 loss and A3A expression levels, however only a couple lines express A3A and this is best addressed within the same cell line background, as there may be many other modulators of the interaction with SMC5/6 loss.

We are not surprised at the lack of correlation between A3A expression and SMC5/6 loss in DepMap, and this is consistent with our data in Figure 4 which show that combined A3A activity and SMC5/6 dysfunction is incompatible with viable human tumors. We appreciate the reviewer's query about the effect of endogenous APOBEC3A on SMC5/6 dependence, which we have now addressed experimentally.

First, we used the NCI-H2347 lung cancer cell line that expresses APOBEC3A at low levels which increases substantially with type I interferon treatment. We found that replication tracts are longer upon IFN treatment and decrease when IFN is combined with depletion of SMC5 (by siRNA). These data are added to Fig. 6i and new Supp Fig. 5. Interestingly, a recent study showed that interferon treatment promoted replication fork acceleration in cancer cell lines (PMID 32597933 Raso, et al JCB 2020). While that paper attributed the effect to ISG15, we now show that SMC5 depletion abrogates increased tract length even in the presence of IFN. These data reflect a pattern similar to that observed in cells with dox-inducible APOBEC3A, suggesting that APOBEC3A upregulation by IFN plays a role in altering replication dynamics. We have included discussion in the manuscript.

Second, we used the PC9 NSCLC cell line which expresses APOBEC3A and a isogenic line in which APOBEC3A was knocked out by CRISPR. We obtained these cells from Aaron Hata's lab which were characterized in their recent publication (Isozaki, et al. PMID: 37407818) and upregulate endogenous APOBEC3A upon EGFR inhibition. We treated PC9 WT and A3Ako cells with low dose EGFRi (0.05uM osimertinib). Cells were transfected with siRNA targeting SMC5 or non-targeting controls. Our preliminary studies found that the combination of endogenous APOBEC3A and SMC5 depletion led to decreased viability. These data support the hypothesis that synthetic lethality occurs in the context of endogenous APOBEC3A. This experimental system is being used in our lab to study the effects of APOBEC3A on drug sensitivity and resistance, and data will be included in a subsequent manuscript.

3) Nearly all of the validation experiments in the paper make use of a single shRNA for SMC5 from what I can gather. Have any of the key findings been complemented or replicated with an additional shRNA?

The reviewer makes an important point about reproducibility. We relied on four different cell lines with different methods of SMC5/6 depletion (shRNA, siRNA, degron tag) to ensure that validations were reproducible. We have now also added a second shRNA targeting SMC5 to key experiments in Figure 2 and Figure S2.

4) Given the role of SMC5/6 in promoting ATR signaling, is SMC5/6 depletion epistatic with ATR inhibition in the context of A3A expression? In other words, does ATR inhibition bypass SMC5/6 activity?

This is an interesting question that was raised by multiple reviewers. The literature suggests that in yeast SMC5/6 may elicit ATR activation (Khan, et al 2022) thus one hypothesis would be that SMC5/6 loss and ATR inhibition are epistatic in the response to replication damage. Alternatively, SMC5/6 and ATR effect distinct pathways to prevent or repair APOBEC3A-mediated damage to replicating DNA. We have added new data (Fig. 2g, Fig. S2i,j) that supports the latter hypothesis. To summarize, we do not see a change in ATR signaling upon SMC5/6 depletion, demonstrating that in these cancer cells SMC5/6 loss does not impact ATR activity. Additionally, we observe compounded lethality with SMC5/6 depletion and ATR inhibition in A3A-expressing cells. These data suggest that SMC5/6 and ATR are not epistatic in the response to APOBEC3A-induced DNA damage.

5) In Figure 5 they look at γ H2AX foci in all cells and cyclin A+ cells. There seems to be an increase in both populations (Fig S4b) not just cyclin A+ cells. Can the authors comment on this and how it would fit into their model. The γ H2AX foci are also very difficult to see in their representative images.

The reviewer raises an interesting question which prompted us to reanalyze the IF imaging of cyclin A-negative cells. We found no statistically significant difference between dox-treated shCtrl v. shSMC5 conditions ($p=0.2637$). Statistics have been added to Fig. S4b. That said, prior studies have shown accumulation of uracil in cells arrested in G1 phase, so it is likely that APOBEC3A acts to some extent on non-replicating DNA, which may require SMC5/6. Also possible is that under-replicated or damaged regions of the genome that persist through mitosis are supported by SMC5/6, and in the absence of SMC5/6 this damage is evident in G1. At the reviewer's suggestion, we replaced several imaging panels to improve visualization of γ H2AX foci in Fig. 5a.

Minor comments

1) The authors propose multiple times that the SMC5/6-A3A interaction complex could be used as a therapeutic vulnerability in cells expressing A3A or in cells with SMC5/6 mutations. Most cell lines tested appear to require intact SMC5/6 for DNA replication and DNA repair (Payne et al. 2014; van der Crabben et al. 2016; Venegas et al. 2020; Grange et al. 2022; Zhu et al. 2023; DepMap). It would be helpful to elaborate more on how this could be put to use in a therapeutic context.

We agree with the reviewer that therapeutic targeting of the entire complex or even one subunit would likely generate substantial toxicity. Ideally, a specific function of the complex would be targeted. For example, the dsDNA binding function which appears to be important for preventing A3A damage. More structural/biochemical work would need to be done to achieve this specific targeting goal.

2) I am not sure the journal policy on this point, but none of the western blots have any molecular weight markers, which I suggest correcting.

We have added molecular weight markers to all blots probed with endogenous antibodies and those with multiple bands. We have additionally submitted source material which includes molecular weight markers for all uncropped blots.

3) In Figures 1, 2 and S2 is there a specific reason for estimating cell viability using three different staining protocols Calcein AM, WST8 and CFSE respectively across THP1, K562 and Jurkat (all suspension) cell lines?

Among the cell lines used, we find that different viability tests are better suited to each cell line depending on rate of growth, confluency, and toxicity of assay reagents.

4) In Supplementary figure S2b with the western blot of SMC5/6 degradation in HCT116 the A3A is HA tagged and is blotted for the HA tag. Why are there two bands for the HA tag?

There are two isoforms of APOBEC3A due to a conserved Kozak sequence at the methionine in position 13 (M13). Prior publications have reported these two APOBEC3A isoforms (PMID 23977391, 20615867). The two bands represent these two isoforms which are evident in almost every blot of either ectopic or endogenous APOBEC3A. Different constructs and different percent gels cause more pronounced visualization of the double band.

5) Previous reports like Venegas et al 2020, Cell reports demonstrated induction in DNA damage (γH2AX) signal upon loss of SMC5/6. In this manuscript, the authors do not see any clear increase in γH2AX after SMC5 silencing, particularly in the leukemia cell lines (THP1 and K562). Similarly, A3A expression by itself does not appear to induce γH2AX or the olive movement significantly as shown previously (Wörmann et al., 2021 Nature cancer). It might be relevant for authors to discuss this, is this a cell line specific effect? Due to timing of depletion or A3A induction?

We do find that A3A alone, when expressed at high enough levels, causes an increase in DNA breaks (by COMET or γH2AX staining). This is shown to some extent in K562, Jurkat, and HCT116 cells with inducible A3A (Fig 2, S2). However, we use inducible systems to enable a low enough level of A3A expression that cells are not incurring substantial DNA damage at the dose of dox we use.

Regarding the loss of SMC5 leading to DNA damage, we do find an increase over baseline in K562, Jurkat, and HCT116 cells (see Fig 2d, S2f, S2g, 5c, e, h). We suspect that the use of partial/transient depletion methods (RNAi/degron tag) mitigates substantial damage caused by SMC5/6 loss. Indeed, we previously knocked out SMC5 by CRISPR and not only had difficulty recovering clones but also found the cells were not able to be passaged beyond 3-4 weeks. Thus, we turned to alternative depletion methods which resulted in less toxicity. These dose-dependent effects are included in the Results section (pages 7-8).

6) In the results the section titled- "A3A catalytic activity is required for synthetic lethality with SMC5/6 loss", authors mention- "Cells expressing APOBEC3A with SMC5/6 depletion had no differences in proliferation, γ H2AX levels, or DSB quantity (Fig2F and Fig S3c-d)." While I presume the results would be similar, they have only tested the effects of SMC5 silencing and not SMC6.

This has been corrected.

7) The authors mention-"Importantly, SMC5/6 depletion did not cause increased APOBEC3A deaminase activity (Fig S3a)." This data appears to show ~50% reduced A3A activity upon SMC5 silencing. If this reduction is statistically significant and reproducible it might be interesting to discuss the potential molecular pathway at play causing SMC5 silencing to reduce A3A activity.

We found a modest decrease in APOBEC3A activity in K562 cells depleted of SMC5, and based on the reviewer's query, we assessed whether SMC5/6 depletion also correlates with decreased APOBEC3A activity in HCT116 cells. We find a slight but not statistically significant decrease in APOBEC3A activity when IAA is used to deplete AID-tagged SMC5/6 components.

While this may represent a negative regulatory effect of SMC5/6 on A3A activity, we do not have enough data to draw conclusions and prefer not to speculate within the manuscript. This is an interesting finding that the reviewer raises, and perhaps a subject for more investigation in our lab. Regardless, a decrease in APOBEC3A activity only highlights the potency of synergistic lethality in SMC5/6 depleted cells.

8) The authors tested the requirement of DNA binding activity of the SMC5/6 complex for its synthetic lethal relationship with A3A in yeast. They state "These data demonstrate that a fundamental feature of SMC5/6 in binding dsDNA that is conserved from yeast to human is responsible for supporting cell viability when APOBEC3A is active." As stated, this implies that this is true in both organisms but the requirement of DNA binding for the synthetic interaction has not been directly tested in human cells.

We thank the reviewer for pointing out this oversight. We have adjusted the language for clarity and the statement now reads: *These data demonstrate that SMC5/6 DNA binding is critical for cell growth when APOBEC3A is active in yeast. Given that DNA binding is a fundamental feature of SMC5/6, an extrapolation of this result is that this activity also protects human cells from genotoxic effects of APOBEC3A.*

9) Authors discuss that SMC5/6 may protect ssDNA gaps through HDR. Are HDR genes enriched in the screen to support this idea? Have the authors performed HDR assays, like the traffic light reporter assay to show that SMC5/6 affect HDR or has this been reported elsewhere?

Yes, in fact DSB Repair was a top GO term (see Fig 1b) from the screen. HDR factors, namely RAD51 and RAD51 paralogues, were significantly negatively selected. We anticipate that these factors are important for replication fork protection/repair when APOBEC3A is active, which is a current direction in our lab. The reported roles of SMC5/6 in HDR relate to resolution of recombination intermediates. We have not done traffic light reporter or similar assays to evaluate how SMC5/6 impacts HDR efficiency, but discuss the published data regarding SMC5/6 in HDR on page 18.

10) The authors report a novel CRISPR screen analysis pipeline. If there is specific code used that goes beyond what is published for the individual tools, it should be provided. It is unclear how these tools are being integrated from the explanation.

We have uploaded the custom code described in Fig. S1c to github.com/khayer/CRISPRkat and added this information to the Methods section.

Referee #2:

In this manuscript, O'Leary et al. identified the SMC5/6 complex as essential for protecting cells expressing active APOBEC3A (A3A). The authors elegantly demonstrated that cells become more sensitive to A3A-induced DNA damage when depleted of SMC5. Moreover, cells lacking SMC5 and expressing A3A exhibited an increase in DNA damage during replication. Interestingly, A3A overexpression led to an acceleration in replication speed dependent on both SMC5 and PrimPol. From this study, the authors propose that SMC5/6 represents a potential therapeutic vulnerability in tumors with active A3A.

The manuscript is very well written, and the experiments are well executed with appropriate controls, including the use of multiple cell lines and catalytic mutants. However, some additional experiments are needed to demonstrate the specificity of the synthetic lethal interaction between A3A activity and loss of SMC5/6. Indeed, it is unclear to the reviewer whether the increase in cell death and DNA damage caused by

the loss of SMC5/6 occurs only in A3A-expressing cells or also in the presence of different types of DNA damage. Data showing the specificity to A3A of this synthetic lethal interaction would increase the impact of the result presented in this manuscript.

Major comments:

A previous publication reported that A3A overexpression in cells does not slow down the replication fork (PMID: 38241374). Another paper showed that A3A-induced abasic sites slow down replication elongation (PMID: 32492421). How do the authors reconcile their findings (increase in replication track lengthening following A3A expression) with these previous studies showing opposite and conflicting results?

We appreciate the reviewer's query and were also surprised by our results initially given the previous publication. Regarding the publication from the Cortez lab (PMID 32492421), we found that upon APOBEC3A expression replication tracts initially become shorter and then elongate. These findings are consistent with what the Cortez lab showed, in that they used a very brief nuclear APOBEC3A expression (30min-4h) prior to evaluating replication tracts. In that time frame, we also find that tracts become shorter. After 24h, we find that tracts elongate which is consistent with a PrimPol-mediated response. We have included these data in the revised manuscript (Fig. 6b).

Regarding the publication from the Zou lab, please see response to Referee 1 above.

In the discussion, it was speculated that the role of PrimPol in A3A-expressing cells is adaptive over time. Do the authors provide evidence supporting this model? It would be beneficial in reconciling the discrepancy with another study (PMID: 32492421) that shows A3A expression slows down replication in the short term.

Yes, please see response above and new data in Fig. 6b.

Does the increase in cell death, DNA damage levels, and replication speed after A3A expression and depletion of SMC5 depend on the formation of abasic sites caused by UNG2?

This is an excellent question which was addressed by the paper from Kawale and colleagues (PMID: 38241374) in which they show that depletion of UNG2 leads to decreased length of replication tracts in cells that express APOBEC3A.

Based on the reviewer's query, we asked whether UNG inhibition would mitigate genotoxicity caused by APOBEC3A in cells with SMC5/6 depletion. We used a uracil glycosylase inhibitor (UGI) and in preliminary studies we found that UGI did not impact replication tract length in APOBEC3A-expressing cells. We also found that UGI did not impact DNA damage or viability of cells in which SMC5/6 is depleted and APOBEC3A is active. This is a perplexing result, since UGI has been previously shown to prevent APOBEC3A-mediated DNA damage (which is replicated in our preliminary data), thus we would expect that UGI would rescue the phenotype of synthetic lethality with

SMC5/6 depletion. It is possible that PrimPol generates gaps in response to uracil, rather than just abasic sites, and therefore ssDNA gaps occur regardless of uracil excision. A recent pre-print from the lab of Lee Zou supports these findings by showing that uracil accumulation leads to PrimPol-dependent ssDNA gaps (Saxena, et al. Unprocessed genomic uracil as a source of DNA replication stress in cancer cells. Biorxiv. Feb 5, 2024). These unreconciled results represent an open question and one we can further interrogate. We hope the reviewer understands this situation and agrees with our approach.

It is unclear how the increase in cell death and DNA damage in cells depleted for SMC5 is specific to DNA damage caused by A3A. Does overexpression of A3B lead to similar phenotypes?

We investigated the possibility that A3B also synergizes with loss of SMC5/6 by generating cell lines with dox-inducible APOBEC3B. Despite similar levels of A3B expression, we found no proliferative defect and no increase in γ H2AX in SMC5-depleted cells relative to controls. We interpret these findings to mean that the synergistic lethality is specific to APOBEC3A among the cancer-associated APOBEC3 enzymes and have added the data to Fig. S3e-g.

Furthermore, does SMC5/6 protect cells against various types of DNA damage that impact replication forks, particularly those involving PrimPol? Or is this protective function specific to DNA damage resulting from cytosine deamination?

This is a really interesting question and one that we are keenly interested in. In particular, we are curious whether mutagens that generate ssDNA gaps during replication or elsewhere in the genome require SMC5/6 to prevent DNA breaks. We have added this future direction to the discussion on page 19. We are pursuing this question experimentally and hope to provide answers in a subsequent paper, but for this manuscript we focus on the specific interaction and mechanism of cytosine deaminase-induced mutagenesis and SMC5/6.

Minor comments:

In Figure S2b, there are two bands for SMC6. Is the bottom band corresponding to an SMC6 allele untagged with mAID?

Both SMC6 alleles have mAID tags in the HCT116 cell line. The bottom band on the SMC6 blot is nonspecific and we have indicated this in the Fig S2 legend.

Bar graphs should show individual data points as dot plots.

All bar graphs represent at least n=3, so we used the SD or SEM bars to represent the spectrum of results.

Can the authors comment on why they did a 1:1 ratio of lentivirus:cell for their CRISPR

screen? In pooled CRISPR screens, cells are usually transduced at a relatively low MOI, often between 0.3 and 0.5 to ensure that only a few cells receive more than one gRNA simultaneously.

The reviewer is correct, the language used in the original submission was misleading and has been corrected in the revised version.

Suggestions to improve the study:

A previous study by the authors showed that cells expressing A3A are more sensitive to ATRi. Are ATR inhibition and SMC5/6 depletion epistatic and non-epistatic? It would be interesting to determine whether A3A-expressing cells depleted of SMC5/6 show increased sensitivity to ATRi, or if this depletion leads to no further sensitivity.

We appreciate the reviewer's suggestion and have done additional experiments to address this question. Details of our findings are in the response to Reviewer 1 above and in new Fig. 2g and Fig. S2i,j.

Referee #3:

In the present study by O'Leary et al, the authors conducted a genetic screen to identify cellular factors exhibiting synthetic lethality with the mutagenic activity caused by APOBEC3A expression. As expected, they found that cells expressing APOBEC3A exhibited synthetic lethality with genes involved in DNA damage repair and chromosome organization. In particular, they found that several subunits of the SMC complex, Smc5/6, appeared on their screen. They subsequently confirmed that depletion of SMC5/6 in cells expressing APOBEC3A causes decreased viability, highlighting the synthetic lethal interaction.

Analysis of tumor data from, further supported the genetic interaction as tumors with mutations in SMC5/6 genes lacked APOBEC3A-induced mutational signatures, indicating an incompatibility between SMC5/6 dysfunction and APOBEC3A activity in viable tumors.

Further investigation revealed that APOBEC3A-induced DNA damage occurred primarily during DNA replication and caused high levels of damage exacerbated by SMC5/6 loss. The authors used DNA fiber spreading to analyse replication tract lengths and found that APOBEC3A expression causes elongated replication tracts, dependent on the DNA primase-polymerase PrimPol and Smc5/6.

Overall, this is a well-executed study with conclusions that are well supported by the data and it highlights the importance of SMC5/6 in protecting cells from APOBEC3A-induced genotoxicity. Moreover, the replication experiments provide some glimpses of the potential mechanisms likely related to replication fork stabilisation in response to lesions.

Specific comments

The main criticism relates to the yeast experiments and the implication that Smc5/6's role in repairing APOBEC3A lesions relies on its ability to bind dsDNA. I think the authors should be cautious because the complex is known to bind both ssDNA and dsDNA and presently molecular mechanisms for fork stabilisation/repair are unclear, and APOBEC3A-induced lesion occur on ssDNA. The authors should discuss this.

We agree with the reviewer and have now added in the Discussion: *Future studies should determine whether Smc5/6 binding to ssDNA, dsDNA, or ss-dsDNA junctions is key for protecting cells from APOBEC3A-induced lesions.*

Further, we have modified the statement in the Result section by not being narrowly focused on dsDNA. The modified statement is: *These data demonstrate that SMC5/6 DNA binding is critical for cell growth when APOBEC3A is active in yeast.*

Dr. Abby M Green
Washington University
Pediatrics
425 S. Euclid Ave.
MPRB 5105
St. Louis, MO 63110

8th May 2024

Re: EMBOJ-2024-116747R
The SMC5/6 complex prevents genotoxicity upon APOBEC3A-mediated replication stress

Dear Abby,

Thank you for submitting your revised manuscript to The EMBO Journal. Two of the original referees have now looked at it again and were generally satisfied with your revisions. We shall be therefore be happy to accept the study after a final round of minor revision, allowing you to presentationally incorporate answers to remain queries of referee 2, and to address the following editorial issues:

- Please reduce the number of keywords on the first manuscript page to 5, ideally choosing general terms over more specific ones
- Please rename the conflict of interest statement into "Disclosure and competing interests statement" as specified in our Guide to Authors.
- As we are switching from a free-text author contribution statement towards a more formal statement based on Contributor Role Taxonomy (CRediT) terms, please remove the present Author Contribution section and instead specify each author's contribution(s) directly in the Author Information page of our submission system during upload of the final manuscript. See <https://casrai.org/credit/> for more information.
- Please refer to the BioImage Archive deposition in the Data Availability section of the text (including accession number and a hyperlink to the database). Likewise, please update the Source Data Checklist (free text box that currently refers to uploading issues) with the access information for the deposited data. Finally, the reference to "Supplemental Table" in the Source Data Checklist should be adjusted (Dataset EV1?).
- Please revise Figure 1D: Firstly, since 'Par.' and 'shSMC5' lanes are not adjacent to each other in the Source Data image, there needs to be a clearly visible separation line in all rows of the final figure, to avoid misrepresentation. Secondly, it appears that the 'Tubulin' bands in the 'Par' lanes really originate from the 'shCTRL' lanes - this has to be corrected. Finally, the bands in the 'A3A-HA' row do not clearly correspond to those in the Source Data - should they stem from a different exposure, the latter would need to be included in the Source Data as well; otherwise, the row should be reassembled with the data from the experiment included in the Source Data.
- Since Fig 2G shows data from only two replicates, plotting of means with errors bars cannot be applied here. Please redraw this panel, plotting individual data points (with mean), and avoid misleading connections via smoothed curves.
- Finally, please provide suggestions for a short 'blurb' text prefacing and summing up the conceptual aspect of the study in two sentences (max. 250 characters), followed by 3-5 one-sentence 'bullet points' with brief factual statements of key results of the paper; they will form the basis of an editor-written 'Synopsis' accompanying the online version of the article. Please also upload a synopsis image, which can be used as a "visual title" for the synopsis section of your paper. The image (which could be simply based on Fig 7D) should be in PNG or JPG format, and please make sure that it remains in the modest dimensions of (exactly) 550 pixels wide and 300-600 pixels high.

I am therefore returning the manuscript to you for a final round of minor revision, to allow you to make these adjustments and upload all modified files. Once we will have received them, we should hopefully be ready to swiftly proceed with formal acceptance and production of the manuscript. Please do not hesitate to contact me in case anything should be unclear.

With kind regards,

Hartmut

*** PLEASE NOTE: All revised manuscripts are subject to initial checks for completeness and adherence to our formatting guidelines. Revisions may be returned to the authors and delayed in their editorial re-evaluation if they fail to comply to the following requirements (see also our Guide to Authors for further information):

9) Digital image enhancement is acceptable practice, as long as it accurately represents the original data and conforms to community standards. If a figure has been subjected to significant electronic manipulation, this must be clearly noted in the figure legend and/or the 'Materials and Methods' section. The editors reserve the right to request original versions of figures and the original images that were used to assemble the figure. Finally, we generally encourage uploading of numerical as well as gel/blot image source data; for details see: embopress.org/page/journal/14602075/authorguide#sourcedata

At EMBO Press, we ask authors to provide source data for the main manuscript figures. Our source data coordinator will contact you to discuss which figure panels we would need source data for and will also provide you with helpful tips on how to upload and organize the files.

In the interest of ensuring the conceptual advance provided by the work, we recommend submitting a revision within 3 months (6th Aug 2024). Please discuss the revision progress ahead of this time with the editor if you require more time to complete the revisions. Use the link below to submit your revision:

Link Not Available

Referee #1:

A3A expression causes DNA damage and mutagenesis across many cancer types and generates a characteristic mutational signature that can be identified. There is significant interest in both how to exploit this signature for therapeutic gain, as well as how cells tolerate A3A expression and its biological effects. The manuscript shows that A3A generates gaps via PrimPol and demonstrates a novel role for SMC5/6 in the process. Further they establish a synthetic lethal relationship between A3A and SMC5/6 that is reflected in the relationship of cancer mutational signatures and SMC5/6 inactivation in cancer. The data will be of broad interest to the genome integrity, APOBEC and cancer fields.

The authors have adequately addressed all major concerns. While some outstanding questions remain, they are outside of the scope of the current manuscript.

Referee #2:

The authors have answered most of my questions, yet two of my primary concerns still need to be addressed. Both questions can be easily answered.

- I couldn't locate the new data on UNG2 mentioned in the rebuttal within the manuscript. Did the authors forget to include them? It is unclear how the experiments using UGI were performed. How did the authors confirm the effectiveness of UNG2 inhibition by UGI? Considering the potential limitations of transient transfection, it remains uncertain if UGI was uniformly expressed in every cell. or were cells stably expressing UGI? A strategy using siRNA targeting UNG2, as described by Kawale and colleagues, may be more suitable (easier) to address this question. This is important since Kawake reported that A3A triggers PrimPol-mediated ssDNA gaps through UNG-generated abasic sites, and the proposed model in this manuscript suggests that SMC5/6 acts downstream of PrimPol.

- My second question was related to determining if SMC5/6 protects cells against various types of DNA damage that impact replication forks, particularly those involving PrimPol. I asked this question because the data shown in Fig 7 does not fully support the proposed model claiming that SMC5/6 acts downstream of PrimPol. Therefore, if SMC5/6 acts after PrimPol, other types of DNA damage leading to PrimPol-mediated ssDNA gaps should be more sensitive to SMC5/6 depletion. Additionally, does PrimPol knockdown prevent cell death caused by SMC5/6 depletion in A3A-expressing cells?

- Bar graphs should show individual data points. This has become the new standard in biological science-related manuscripts. [EDITOR'S NOTE: as discussed, EMBO Journal continues to allow showing bar diagrams with error bars as long as $N > 2$]

Thanks again to both Referees for reviewing and commenting on the manuscript. We appreciate your time and advice, and are glad to have answered most questions. Remaining queries are addressed below.

Referee #1:

A3A expression causes DNA damage and mutagenesis across many cancer types and generates a characteristic mutational signature that can be identified. There is significant interest in both how to exploit this signature for therapeutic gain, as well as how cells tolerate A3A expression and its biological effects. The manuscript shows that A3A generates gaps via PrimPol and demonstrates a novel role for SMC5/6 in the process. Further they establish a synthetic lethal relationship between A3A and SMC5/6 that is reflected in the relationship of cancer mutational signatures and SMC5/6 inactivation in cancer. The data will be of broad interest to the genome integrity, APOBEC and cancer fields.

The authors have adequately addressed all major concerns. While some outstanding questions remain, they are outside of the scope of the current manuscript.

Referee #2:

The authors have answered most of my questions, yet two of my primary concerns still need to be addressed. Both questions can be easily answered.

- I couldn't locate the new data on UNG2 mentioned in the rebuttal within the manuscript. Did the authors forget to include them? It is unclear how the experiments using UGI were performed. How did the authors confirm the effectiveness of UNG2 inhibition by UGI? Considering the potential limitations of transient transfection, it remains uncertain if UGI was uniformly expressed in every cell. or were cells stably expressing UGI? A strategy using siRNA targeting UNG2, as described by Kawale and colleagues, may be more suitable (easier) to address this question. This is important since Kawale reported that A3A triggers PrimPol-mediated ssDNA gaps through UNG-generated abasic sites, and the proposed model in this manuscript suggests that SMC5/6 acts downstream of PrimPol.

We did not include experiments using UGI in the manuscript, as they are only preliminary data. We have included results below along with experimental description. We maintain that the DNA lesion to which PrimPol responds may be either an abasic site (as reported by Kawale et al) or a uracil (as suggested by the recent preprint from Lee Zou's lab by Saxena, et al). While Kawale's data suggest the former, our preliminary results using UGI to inhibit UNG2 suggest the latter. This open question will be pursued in our lab. Regardless of the lesion to which PrimPol is

responding, we think the most likely model to explain our data puts SMC5/6 functioning after PrimPol generates a ssDNA gap.

- My second question was related to determining if SMC5/6 protects cells against various types of DNA damage that impact replication forks, particularly those involving PrimPol. I asked this question because the data shown in Fig 7 does not fully support the proposed model claiming that SMC5/6 acts downstream of PrimPol. Therefore, if SMC5/6 acts after PrimPol, other types of DNA damage leading to PrimPol-mediated ssDNA gaps should be more sensitive to SMC5/6 depletion. Additionally, does PrimPol knockdown prevent cell death caused by SMC5/6 depletion in A3A-expressing cells?

Our hypothesis is that PrimPol-mediated gaps generate ssDNA-dsDNA junctions which require SMC5/6 to stabilize replication forks. We have come to this hypothesis based on data in Figure 7c which shows that PrimPol depletion mitigates DNA damage caused by the combination of SMC5/6 depletion and APOBEC3A expression. We did not directly assess viability, but instead use γ H2AX as a proxy for genotoxicity. Therefore, we agree with the Reviewer that mutagens beyond APOBEC3A that generate ssDNA gaps should require SMC5/6 to prevent replication fork collapse or DSB generation. We have not experimentally addressed this question for the current study, but agree that it is an important question to answer in the future. Importantly, we acknowledge that alternative hypotheses are possible as discussed on page 20 of the manuscript.

- Bar graphs should show individual data points. This has become the new standard in biological science-related manuscripts.

[EDITOR'S NOTE: as discussed, EMBO Journal continues to allow showing bar diagrams with error bars as long as $N > 2$]

Dr. Abby M Green
Washington University
Pediatrics
425 S. Euclid Ave.
MPRB 5105
St. Louis, MO 63110

17th May 2024

Re: EMBOJ-2024-116747R1
The SMC5/6 complex prevents genotoxicity upon APOBEC3A-mediated replication stress

Dear Dr. Green,

Thank you for submitting your final revised manuscript for our consideration. I am pleased to inform you that we have now accepted it for publication in The EMBO Journal.

Yours sincerely,

Hartmut Vodermaier
